# A comprehensive map coupling histone modifications with gene regulation in adult dopaminergic and serotonergic neurons

Erik Södersten[1], Konstantinos Toskas[1], Vilma Rraklli[1], Katarina Tiklova[1], Åsa K. Björklund [2], Markus Ringnér [3], Thomas Perlmann[1] & Johan Holmberg[1]

The brain is composed of hundreds of different neuronal subtypes, which largely retain their identity throughout the lifespan of the organism. The mechanisms governing this stability are not fully understood, partly due to the diversity and limited size of clinically relevant neuronal populations, which constitute a technical challenge for analysis. Here, using a strategy that allows for ChIP-seq combined with RNA-seq in small neuronal populations in vivo, we present a comparative analysis of permissive and repressive histone modifications in adult midbrain dopaminergic neurons, raphe nuclei serotonergic neurons, and embryonic neural progenitors. Furthermore, we utilize the map generated by our analysis to show that the transcriptional response of midbrain dopaminergic neurons following 6-OHDA or methamphetamine injection is characterized by increased expression of genes with promoters dually marked by H3K4me3/H3K27me3. Our study provides an in vivo genome-wide analysis of permissive/repressive histone modifications coupled to gene expression in these rare neuronal subtypes.

[1] Department of Cell and Molecular Biology, Ludwig Institute for Cancer Research, Karolinska Institutet, Nobels väg 3, 171 77 Stockholm, Sweden.
[2] Department for Cell and Molecular Biology, National Bioinformatics Infrastructure Sweden, Science for Life Laboratory, Uppsala University, 751 24 Uppsala, Sweden. [3] Department of Biology, National Bioinformatics Infrastructure Sweden, Science for Life Laboratory, Lund University, 223 62 Lund, Sweden. Correspondence and requests for materials should be addressed to E.Söd. (email: erik.sodersten@gmail.com) or to J.H. (email: johan.holmberg@ki.se)

While single-cell RNA-sequencing technology has begun to delineate how neuronal subtypes in the adult brain are defined by unique patterns of gene expression[1], less is known about how stable long-term silencing of alternative lineages and progenitor genes is maintained to preserve subtype identity[2]. Increasing efforts have therefore been directed to understand how aberrant neuronal gene regulation is involved in the etiology of neurodevelopmental, neurodegenerative, and mental disorders[3,4]. Within this context, epigenetic mechanisms that facilitate regulation of chromatin structure, in particular through dynamic modification of histones, have gained much attention[3,4].

A major obstacle for such studies is that many clinically interesting populations exist in numbers far lower than required for crucial analytical methods, such as chromatin immunoprecipitation for histone modifications followed by sequencing (ChIP-seq). Midbrain dopamine-producing neurons (mDA neurons) control vital brain functions but only exist in numbers, nearly 50-fold lower (20,000–30,000 cells per mouse) than those required for a single conventional ChIP-seq analysis[5,6]. Still, a better understanding of mDA-specific gene regulation is required as degeneration of mDA neurons is the hallmark of Parkinson's disease, and altered dopamine transmission has implications for addiction and psychiatric disease. In addition to low cell numbers, the entangled cellular environment of the adult brain represents a barrier for isolation of distinct neuronal populations. Therefore, previous work addressing gene regulatory mechanisms in the adult brain has been hampered by the lack of cell-type resolution[4,7,8].

Several studies describe how the combinatorial distribution of repressive and permissive histone modifications correlates with gene expression[9,10]. Polycomb-repressive complex 2 (PRC2) proteins mediate and maintain cell-type-specific gene repression during development[9] through the enzymatic components EZH1/2, which catalyze deposition of the PRC2-associated repressive histone modifications H3K27me2 and H3K27me3 on regulatory genes[9,11,12]. Notably, subtype restricted loss of H3K27me3 in adult medium spiny neurons (MSNs) leads to derepression of genes associated with other lineages, along with apoptosis genes, eventually leading to neurodegeneration[12]. Importantly, changes in PRC2 activity, as well as alterations in H3K27me3 levels and distribution have been associated with neurodegenerative diseases, including Parkinson's disease[13–15].

H3K9me3 is an additional histone modification associated with repression, also suggested to act as a barrier for cell fate changes[16]. Similar to H3K27me3, deposition of H3K9me3 is cell-type specific and appears to regulate gene expression in several ways, ranging from occurring on large gene clusters in megabase-scale domains to individual enhancers[10,17]. H3K9me3 recruits heterochromatin protein 1 (HP-1) family proteins that in turn have important roles in heterochromatin formation[18]. Notably, HP1β has been shown to be required for proper cortical development[19], implying a role for H3K9me3 in regulation of neuronal gene expression.

To overcome the obstacles related to heterogeneity and limited cell numbers, and investigate the association between histone modifications and subtype-specific gene expression, we adopted an approach that allowed us to generate several genome-wide ChIP-seqs combined with RNA-seq data from sparse neuronal subtypes from single adult mouse brains. Here, we present a genome-wide comparison between permissive and repressive histone modifications correlated with gene expression in neural progenitor cells (NPCs) and two clinically relevant neuronal subtypes in vivo: mDA neurons and raphe nuclei serotonergic neurons (SER neurons). Our study elucidates how silencing of both progenitor genes and genes determining alternative neuronal subtypes in mDA neurons correlates with the distribution of H3K27me3, H3K9me3, and the H3K4me3 modifications. Furthermore, we show that dopaminergic stress-induced gene expression in a mouse model of Parkinson's disease, or after methamphetamine injection, is characterized by derepression of genes with promoter regions dually marked by H3K4me3 and H3K27me3, whereas induction of genes with promoter regions marked by any other combination of H3K27me3 and H3K9me3 occurs less frequently.

Our study provides an in vivo genome-wide analysis of permissive/repressive histone modifications coupled to gene expression in these rare but clinically relevant neuronal subtypes. This strategy can be generalized for identification and functional characterization of additional mechanisms involved in the maintenance of gene expression in other classes of neurons.

## Results

**Purification of neural progenitor and adult neuronal nuclei.** To perform comparative analysis of chromatin state[20] and gene expression between defined populations of mature neurons, we utilized a transgene mouse model carrying floxed alleles for a ribosomal protein fused to mCherry (Rpl10a-mCherry[21]). *Rpl10a-mCherry*[flox/flox] mice were crossed with *DatCreER*[T2][22] or *SERT-cre*[23] mice to induce expression of the Rpl10a-mCherry protein in dopamine- or serotonin-producing neurons, respectively. The *Rpl10a-mCherry* mouse line was originally developed for performing translating ribosome affinity purification experiments[21]. However, partial nuclear localization of the Rpl10a-mCherry fusion protein allows for the separation of nuclei from neurons expressing *Cre* under particular cell-type-specific promoters from the surrounding tissue by fluorescence-activated cell sorting (FACS) of the total ex vivo isolated nuclei[24,25]. The specificity of Rpl10a-mCherry expression was confirmed by co-localization of the dopamine cell-specific marker tyrosine hydroxylase (TH), in coronal midbrain sections of *DatCreER*[T2]-*Rpl10a-mCherry* mice (Fig. 1a). The overlap between mCherry and tryptophan hydroxylase 2 (TPH2), a marker for serotonin-producing neurons was confirmed in hindbrain sections from *SERT-cre-Rpl10a-mCherry* mice (Supplementary Figure 1a). FACS analysis of the total extracted mouse midbrain nuclei from *DatCreER*[T2]-*Rpl10a-mCherry* mice showed a distinct mCherry-positive population not present in wild-type mice (Fig. 1b, Supplementary Figure 2).

To validate that FACS isolated mCherry-positive nuclei from *DatCreER*[T2]-*Rpl10a-mCherry* mice represented a pure mDA population, we performed single-nuclei RNA-seq analysis on sorted nuclei. Out of 98 sequenced nuclei, 89 passed quality control (Methods). Principal component analysis (PCA) of normalized log-expression values of correlated highly variable genes (Methods) revealed three nuclei as outliers that were also distinguished by absent or low expression of key dopaminergic marker genes, including *Th*, *Slc6a3* (dopamine transporter, DAT), and *Ret* (Fig. 1c and Supplementary Figure 3a). We concluded that 86 out of 89 (96.6%) FACS-sorted nuclei from *DatCreER*[T2]-*Rpl10a-mCherry* mice originated from mDA neurons. Analysis of confirmed mDA nuclei revealed that although the population might harbor two or more functional mDA subtypes, such differences were based on differential expression of a low number of genes. Thus, we concluded that the sorted nuclei represented a pure and relatively homogeneous mDA population (Supplementary Figure 3b,c, Methods).

To compare the chromatin state of adult mDA neurons to the state of a corresponding NPC population, we collected NPCs from the developing brain. At E11.5, the transcription factor SOX2 labels NPCs throughout the developing CNS, including

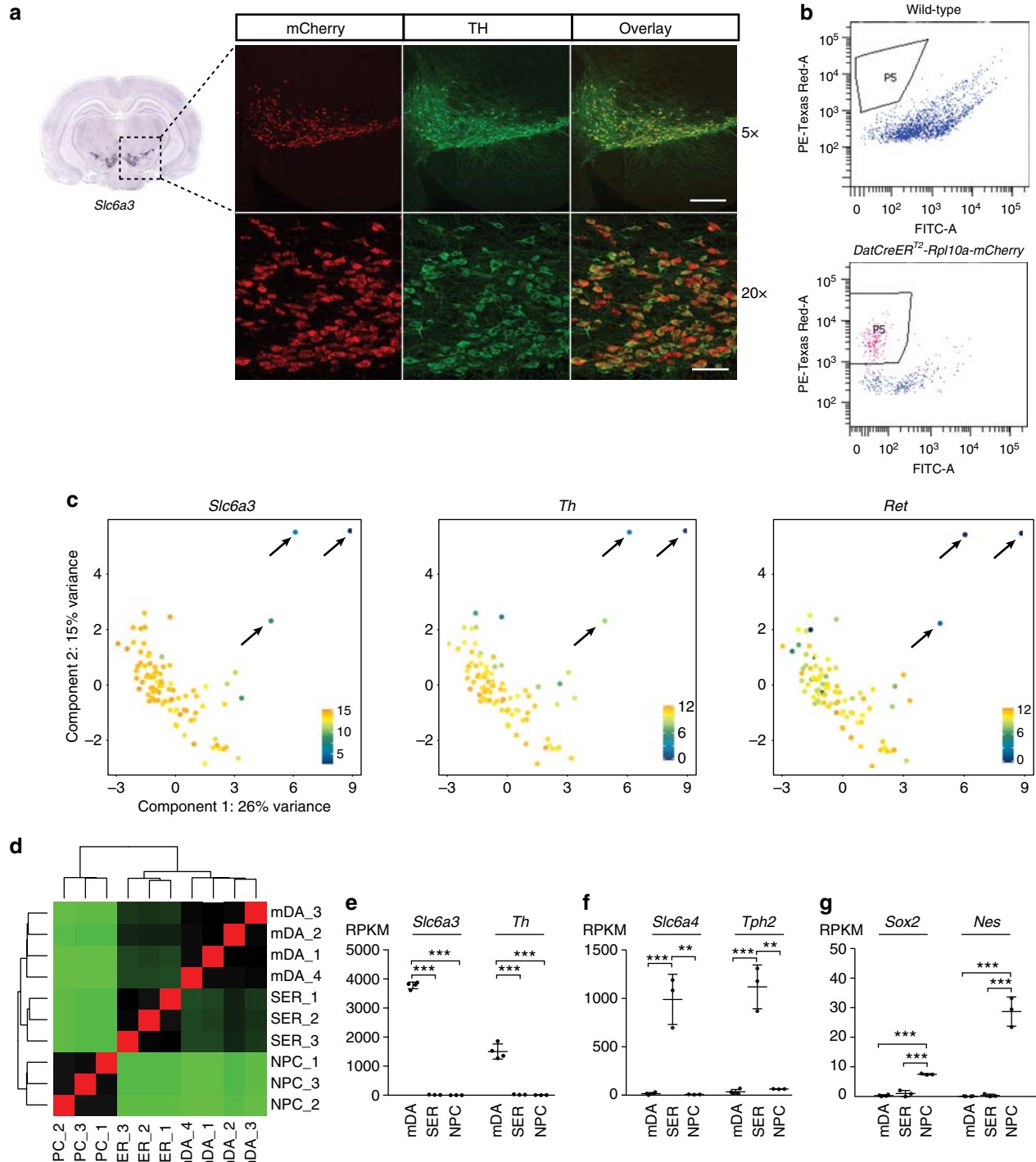

**Fig. 1** Purification of adult neuronal and neural progenitor nuclei. **a** Rpl10a-mCherry expression (red) overlaps TH (green) expression in the ventral midbrain in *DatCreER^T2^-Rpl10a-mCherry* mice. 5x and 20x magnification. To the left is an in situ hybridization for *Slc6a3* taken from the Allen Brain Atlas (mouse. brain-map.org, image credit: Allen Institute), indicating the position of the images. Scale bars: 200 μm (upper panels), 50 μm (lower panels). **b** FACS plots showing a population of nuclei, indicated by "P5," occurring in *DatCreER^T2^-Rpl10a-mCherry* mice but not in wild-type mice. Single-nuclei RNA-seq: **c** PCA-plots constructed from normalized single-nuclei log-expression values of correlated HVGs. Each nucleus is colored according to the expression of *Slc6a3*, *Th*, or *Ret*. Arrows indicate three outlier nuclei identified as originating from non-mDA neurons. A total of 1000 nuclei bulk RNA-seq: **d** Heatmap showing sample-to-sample distances between hierarchically clustered mDA, SER, and NPC RNA-seq libraries. **e** Expression of dopaminergic neuron markers is restricted to mDA nuclei: bars indicate average RPKM ± SD for *Slc6a3* and *Th* as RPKMs in mDA (*n* = 4 mice), SER (*n* = 3 mice), and NPC nuclei (*n* = 3 embryos). **f** Expression of serotonergic markers is restricted to SER nuclei: bars indicate average RPKM ± SD for *Slc6a4* and *Tph2* as RPKMs in mDA, SER, and NPC nuclei. **g** Expression of neural progenitor markers is restricted to NPC nuclei: bars indicate average RPKM ± SD for *Sox2* and *Nes* as RPKMs in mDA, SER, and NPC nuclei. Significance for **e**–**g** according to two-tailed Student's *t*-test assuming equal variances. ***$p \leq 0.001$, **$p \leq 0.01$

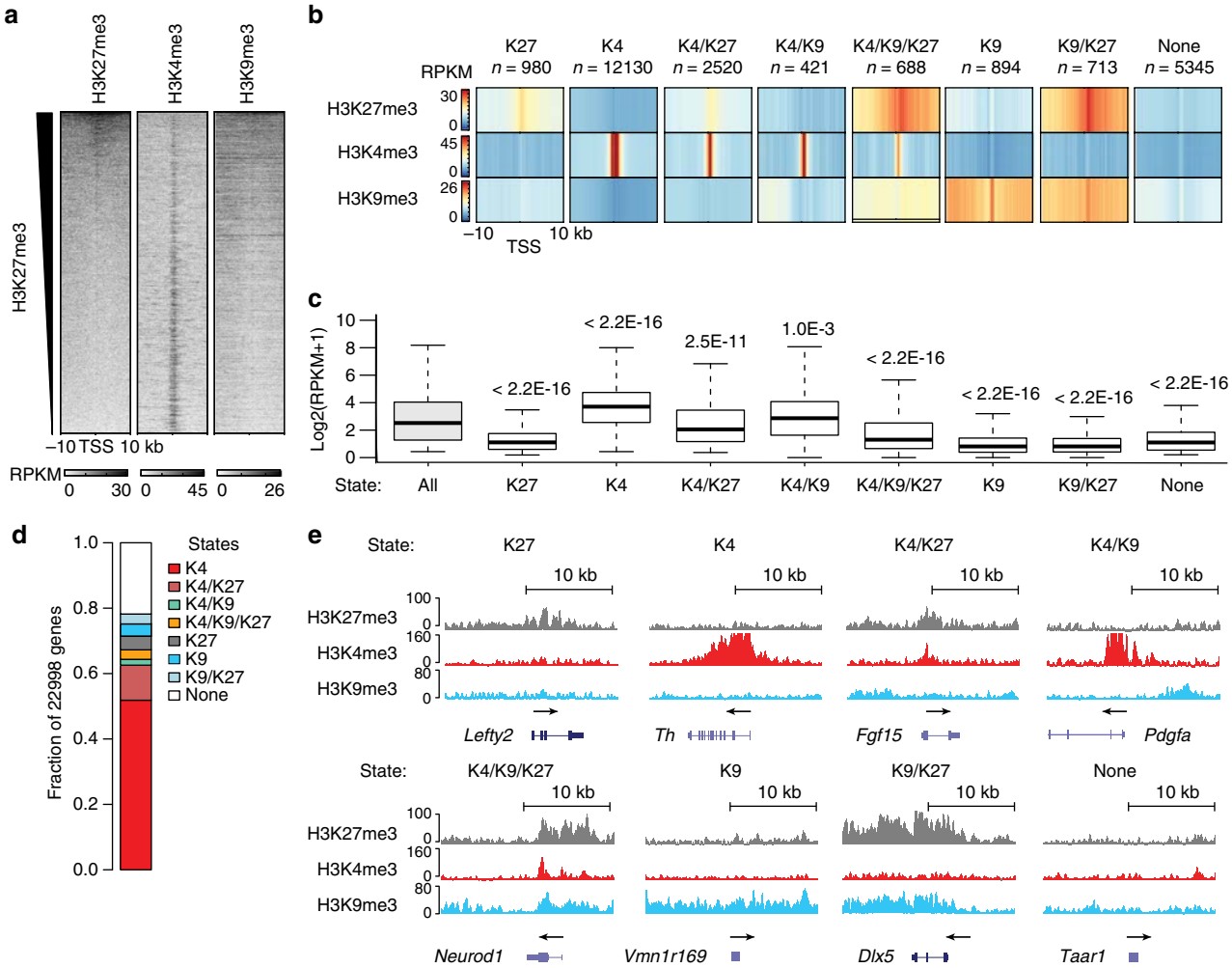

**Fig. 2** Histone modifications and gene expression in adult mDA neurons. **a** H3K27me3 and H3K9me3 abundance around TSS is inversely correlated with H3K4me3 abundance: Heatmaps showing genome-wide abundance of H3K27me3, H3K4me3, and H3K9me3 ± 10 kb around TSS of individual genes in mDA neurons as RPKMs obtained by ChIP-seq. Heatmaps were sorted for descending H3K27me3 abundance. **b** Categorization of TSS regions into chromatin states: Heatmap profiles of average H3K27me3, H3K4me3, and H3K9me3 RPKMs ± 10 kb around TSS of genes per defined chromatin state in mDA neurons (denoted as K27, K4, K4/K27, K4/K9, K4/K9/K27, K9, K9/K27, and None, respectively). **c** Individual chromatin states are associated with different levels of gene expression: box-plots showing the expression levels (log2(RPKM + 1)) of genes in the different chromatin-state categories. The center line is the median, bounds are the 25th and 75th percentiles, and whiskers are ±1.5 IQR. Chromatin states with an average gene expression that is different compared to the global average gene expression are indicated by *p*-values obtained by a two-sided Wilcoxon rank-sum test. **d** Genome-wide relative abundance of ChIP-seq defined chromatin states in promoter regions ±10 kb around TSS in mDA neurons. **e** UCSC Genome browser excerpts (https://genome.ucsc.edu) showing examples of enrichment for H3K27me3 (gray, vertical scale 0–100 RPKM), H3K4me3 (red, vertical scale 0–160 RPKM), and H3K9me3 (blue, vertical scale 0–80 RPKM) at representative genes per chromatin state in mDA neurons. Each plot spans 20 kb

those giving rise to mDA and SER neurons (Supplementary Figure 1b)[26]. We collected nuclei from four *DatCreER^T2-Rpl10a-mCherry* mice (hereby denoted as mDA nuclei), three *SERT-cre-Rpl10a-mCherry* mice (SER nuclei), as well as nuclei from SOX2+ NPCs from three E11.5 embryos (NPC nuclei), using a fluorophore-labeled antibody against SOX2 (Supplementary Figures 2, 4, and 5). Nuclei were collected in batches of 1000, yielding up to seven batches of mDA nuclei and up to five batches of SER nuclei per mouse. mDA, SER, and NPC-nuclei libraries were prepared and RNA-seq performed on one 1000-nuclei batch per mouse. Clustering analysis[27] showed, as expected, that adult neuronal populations (mDA, SER) were more similar to each other compared to NPCs (Fig. 1d). Dopaminergic markers were specifically detected in the mDA nuclei, serotonergic markers were specifically detected in SER nuclei, and markers for NPCs were restricted to NPC nuclei (Fig. 1e–g). We also confirmed that progenitors giving rise to mDA and SER neurons were

represented within the population of sorted NPCs, as the NPC population expressed *Gata3*, a marker for serotonergic progenitors continuously expressed in SER neurons, as well as *Lmx1a*, a marker for dopaminergic progenitors continuously expressed in mDA neurons (Supplementary Figure 1c).

In summary, we isolated pure populations of mDA and SER nuclei based on subtype-specific expression of *Rpl10a-mCherry* in adult mouse brain and SOX2+ e11.5 NPCs by FACS. Purity was confirmed by single-nuclei RNA-seq for mDA nuclei, as well as bulk RNA-seq on sorted mDA, SER, and NPC nuclei.

**Histone modifications and gene expression in mDA neurons.** To investigate how gene expression is reflected by repressive chromatin in adult mDA neurons, we analyzed genome-wide distribution of H3K27me3 and H3K9me3 along with the permissive modification H3K4me3 by ChIP-seq in batches of sorted mDA nuclei. The number of mDA neurons in the adult mouse

brain is less than 30,000[5], and the yield of mDA nuclei was typically 5000–7000 per *DatCreER*[T2]*-Rpl10a-mCherry* mouse after FACS. For standard ChIP-seq experiments, a million cells per antibody are typically used, which would have required us to use up to 200 mice per ChIP. Instead, we turned to a recently developed native ChIP-method suited for analysis of low cell numbers, enabling us to perform ChIP-seq on 1000 sorted nuclei per antibody (ref. [28], Methods). With this approach, 5000 sorted

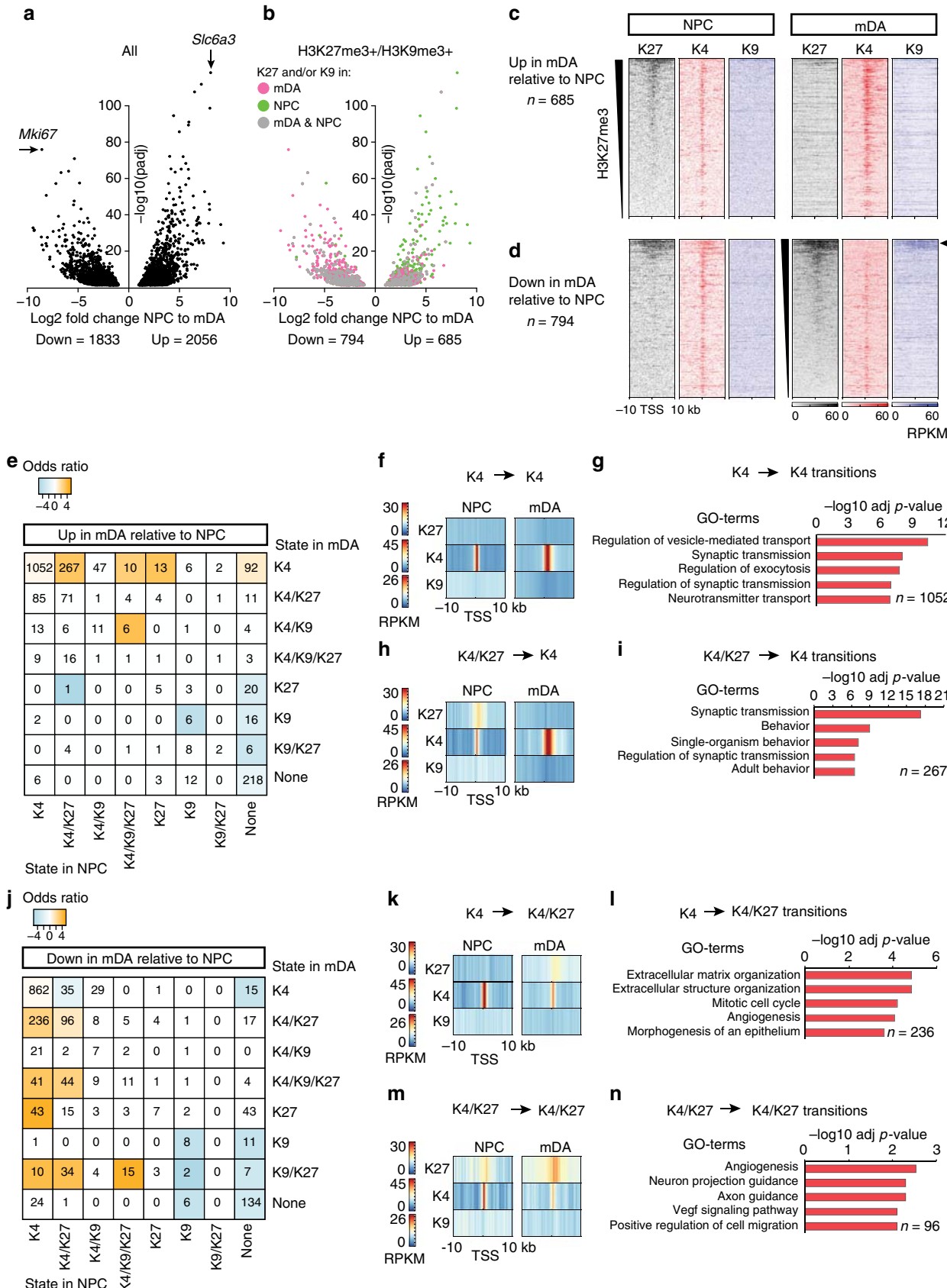

mDA nuclei from a single mouse were sufficient to generate ChIP-seq libraries for H3K27me3, H3K9me3, H3K4me3 and input, as well as a bulk RNA-seq library on 1000 nuclei from the same mouse. ChIP-seq and RNA-seq were performed on at least three mice per condition for the purpose of increased coverage depth and statistical power in the subsequent analyses (Methods).

The abundance of H3K27me3 and H3K9me3 in the promoter region ±10 kb around transcription start sites (TSS) of genes was inversely correlated with enrichment of H3K4me3 (Fig. 2a). To understand how the presence of investigated modifications around TSS related to gene-expression levels, every gene was assigned a chromatin state, based on the combinatorial presence of H3K27me3, H3K4me3, and H3K9me3. With this definition, there were eight chromatin states: "K27," "K4," "K4/K27," "K4/K9," "K4/K9/K27," "K9," "K9/K27," and a chromatin state lacking significant enrichment of any of the analyzed histone modifications, "None". To visualize the distribution of enrichment of each modification within promoter regions, we generated average profiles of read coverage for each chromatin state (Fig. 2b). H3K27me3 enrichment was on average centered at the TSS in the K27 and K4/K27 states, but extended over the entire 20-kb region when co-occurring with H3K9me3 in the K4/K9/K27 and K9/K27 states. Interestingly, the H3K4me3 and H3K27me3 peaks within the K4/K27 state were on average centered at the TSS and highly overlapping. This pattern suggests that genes within the K4/K27 state may contain a subset of genes with bivalent promoters (Fig. 2b,e), as previously reported for other types of adult neurons[12]. We noted that the read density of H3K9me3 was strongly enriched in the K9 and K9/K27 states and less prominent in the K4/K9 and K4/K9/K27 states (Fig. 2b,e). Average gene expression levels per chromatin state were calculated and compared to the total average expression. This revealed that genes belonging to the K4 state displayed a significantly higher average expression, whereas genes belonging to the other states had a lower average expression (Fig. 2c). Co-occurrence of H3K4me3 and repressive modifications generally correlated with a higher level of transcription compared to genes belonging to states with repressive modifications only (Fig. 2c). Genome-wide, the K4 state was the most abundant (Fig. 2b,d). Notably, the examined repressive histone modifications rarely occurred alone without the simultaneous presence of H3K4me3 and/or H3K27me3/H3K9me3 (Fig. 2b,d).

In summary, the distribution of the analyzed histone modifications correlated with gene expression, i.e., genes belonging to chromatin states with increased levels of H3K4me3 were associated with increased levels of expression and vice versa for genes belonging to chromatin states containing the repressive H3K27me3 and H3K9me3 modifications.

**Chromatin-state transitions reflect mDA-neuron development.** To understand how silencing of developmental genes is reflected by histone modification redistribution in mDA neurons, we compared expression data between mDA neurons and NPCs. This showed that 3889 genes were differentially expressed between NPCs and mDA neurons (Fig. 3a). ChIP-seq in NPCs for H3K27me3, H3K4me3, and H3K9me3 revealed that the abundance of H3K27me3 and H3K9me3 within promoter regions ±10 kb surrounding the TSS of genes in NPCs was inversely correlated with abundance of H3K4me3 (Supplementary Figure 6a). As with mDA neurons, promoter regions were classified according to chromatin state (Supplementary Figure 6b). Expression levels of genes in different chromatin states correlated well with the combinatorial presence of permissive and repressive histone modifications (Supplementary Figure 6b–e). A large proportion of differentially expressed genes between NPCs and mDA belonged to a repressive chromatin state in NPCs and in mDA neurons (Fig. 3b). Inspection of promoter regions belonging to H3K9me3- and/or H3K27me3-containing states revealed that genes upregulated in mDA neurons relative to NPCs lost H3K27me3 and gained H3K4me3 (Fig. 3c). Conversely, genes downregulated in mDA relative to NPCs gained H3K27me3 and displayed reduced H3K4me3 around a subset of TSS (Fig. 3d). Notably, promoter regions with the broadest abundance of H3K27me3 in NPCs and mDA neurons gained H3K9me3 upon terminal differentiation into mDA (Fig. 3d, arrow). Thus, gene expression changes between NPCs and mDA neurons correlated well with transitions between specific chromatin states.

To examine the extent of transitions between chromatin states for individual regulated genes, we constructed transition matrices for up and downregulated genes mapped to their chromatin state in NPC and mDA neurons, respectively (Fig. 3e). An odds ratio was computed for enrichment of genes in each of the 64 possible transitions between the eight defined chromatin states in the two cell types, as compared to all genes (Methods). Genes upregulated in mDA relative to NPCs were dominated by TSS belonging to K4 in both NPCs and mDA ($n = 1052$, 51.2%) close to expected proportions (1.18:1, $p = 3E-11$, odds ratio, and Fisher's exact test corrected for multiple comparisons) (Fig. 3e). Analysis of promoter regions for the K4 > K4 transition between NPCs and mDA showed an increased enrichment of H3K4me3 in mDA compared to NPCs as the genes were upregulated (Fig. 3f). Gene ontology (GO) analysis revealed that this class of loci was predominantly represented by genes encoding proteins involved in basic neuronal function, including regulation of vesicle-mediated transport and synaptic transmission (Fig. 3g), e.g., Syn1 (Supplementary Figure 7a, Supplementary Data 1). Another predominant transition was promoter regions marked as K4/K27

**Fig. 3** Chromatin-state transitions reflect mDA development. **a** RNA-seq: Volcano plot showing differential gene expression between NPC and mDA neurons. Differentially expressed genes with an adjusted $p$-value < 0.05 obtained by Wald test are shown. **b** Volcano plot as in **a** where differentially expressed and H3K27me3 and/or H3K9me3-marked genes are colored according to: marked in mDA – pink, marked in NPC – green, and marked in both mDA and NPC – gray. **c** ChIP-seq heatmaps showing the abundance of H3K27me3, H3K4me3, and H3K9me3 as RPKMs ± 10 kb around TSS of genes that were upregulated in mDA and associated with a chromatin state containing H3K27me3 and/or H3K9me3 and **d** genes that were downregulated in mDA and associated with a chromatin state containing H3K27me3 and/or H3K9me3, respectively. The arrow indicates H3K27me3+ genes that gain H3K9me3 in mDA. **e** Chromatin-state transitions between NPC and mDA neurons for upregulated genes in mDA. Colored squares indicate transitions that occurred significantly more often than expected (odds ratio, orange = more than expected, blue = less than expected, $p < 0.05$ according to Fisher's exact test and adjusted for multiple testing). **f** Heatmaps of average H3K27me3, H3K4me3, and H3K9me3 RPKMs ± 10 kb around TSS of genes upregulated from NPC to mDA and K4 in NPC and K4 in mDA. **g** Gene ontology (GO) of the top ($-$log10 (adj $p$-value), Fisher's exact test) biological processes for upregulated genes in mDA that were K4 in NPC and K4 in mDA. **h** Heatmaps as in **f** of genes upregulated in mDA and K4/K27 in NPC and K4 in mDA. **i** GO of the top biological processes for upregulated genes in mDA that were K4/K27 in NPC and K4 in mDA. **j** Chromatin-state transitions between NPC and mDA neurons for downregulated genes in mDA (**e**). **k** Heatmaps as in **f** of genes downregulated in mDA and K4 in NPC and K4/K27 in mDA. **l** GO of the top biological processes for downregulated genes in mDA that were K4 in NPC and K4/K27 in mDA. **m** Heatmaps as in **f** of genes downregulated in mDA, and categorized as K4/K27 in NPC and K4/K27 in mDA. **n** GO of the top biological processes for downregulated genes in mDA that were K4/K27 in NPC and K4/K27 in mDA

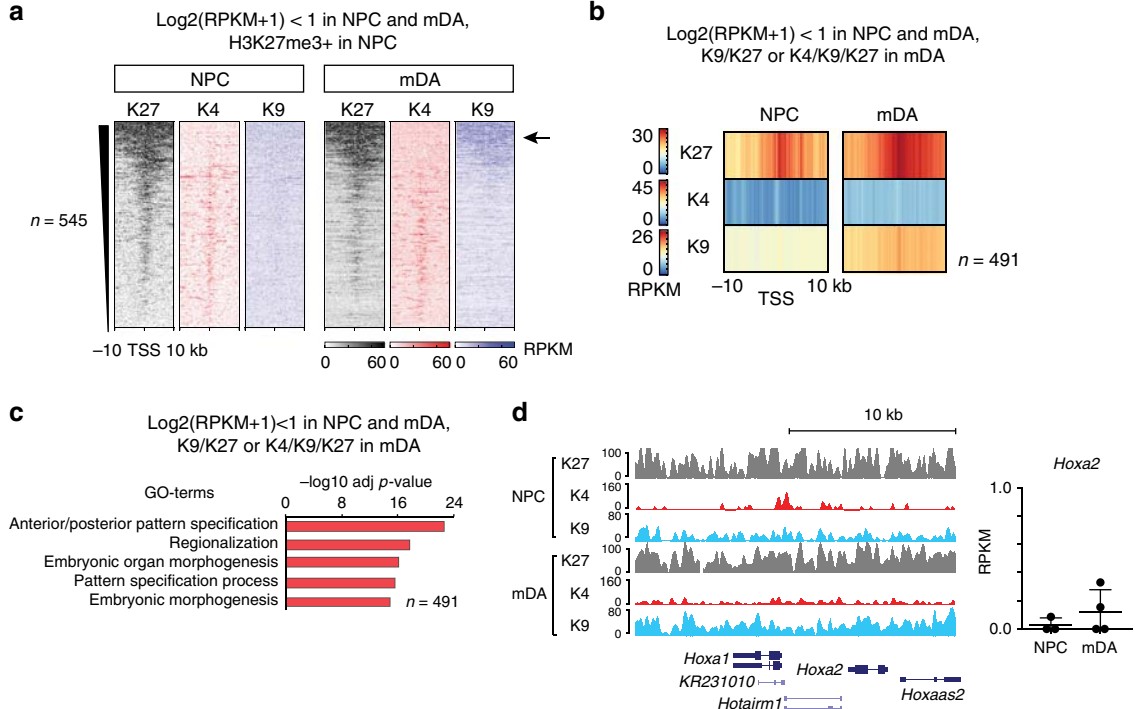

**Fig. 4** Expansion of H3K27me3/H3K9me3 onto silenced genes in mDA neurons. **a** Genes repressed and H3K27me3-marked in both NPC and mDA gain H3K9me3 during terminal differentiation into mDA neurons: ChIP-seq heatmaps showing the abundance of H3K27me3, H3K4me3, and H3K9me3 as RPKMs ± 10 kb around TSS in NPC and mDA of silent (log2(RPKM + 1) < 1) and H3K27me3-marked genes in NPC. The arrow indicates a subset of silent and H3K27me3-marked genes displaying an increase in H3K9me3 in mDA compared to NPC. **b** Heatmap profiles of average H3K27me3, H3K4me3, and H3K9me3 RPKMs ± 10 kb around TSS of silent genes (log2(RPKM + 1) < 1) in NPC and mDA and categorized as K9/K27 or K4/K9/K27 in mDA. **c** Gene ontology (GO) of the top (−log10 (adj p-value), Fisher's exact test) biological processes for silent genes (log2(RPKM + 1) < 1 in NPC and mDA neurons as in **b** that were K9/K27 or K4/K9/K27 in mDA neurons (n = 491). **d** Example of a silent and H3K27me3-marked gene in NPC that gains H3K9me3 in mDA: Left: UCSC Genome browser excerpt showing enrichment for H3K27me3 (gray, vertical scale 0–100 RPKM), H3K4me3 (red, vertical scale 0–160 RPKM), and H3K9me3 (blue, vertical scale 0–80 RPKM) within the *Hoxa*-locus. Right: Bars indicate average RPKM ± SD for *Hoxa2*, which is <1 in NPC and mDA

in NPCs that were resolved into K4 in mDA neurons when the gene was upregulated (n = 267, 13.0%) (Fig. 3e,h). This transition occurred more frequently than expected (3.9:1, p = 3E-91, odds ratio, and Fisher's exact test corrected for multiple comparisons) and contained genes (e.g., *Nr4a2*, *Slc6a3*, and *Drd2*) encoding proteins involved in synaptic transmission in dopamine neuron-specific processes (Supplementary Figure 7b, Supplementary Data 1).

To understand how repression of NPC-specific genes is reflected by the presence of the analyzed histone modifications in mDA neurons, we investigated transitions between chromatin states in NPCs and mDA neurons for genes downregulated in mDA relative to NPC. Similar to the upregulated genes, the largest group of downregulated genes belonged to the K4 state in both NPC and mDA (n = 862, 47.0%), and this transition occurred in amounts similar to the abundance of genes belonging to the K4 state genome-wide (1.09:1, p = 0.004, odds ratio, and Fisher's exact test corrected for multiple comparisons) (Fig. 3j). Promoter regions undergoing K4 > K4 transition and encoding downregulated genes from NPC to mDA showed a decreased enrichment of H3K4me3 in mDA compared to NPCs, contrary to promoter regions of upregulated genes (Supplementary Figure 7e). Many of these loci encoded cell-cycle genes, including *Ccnb1* and *Cdca2* (Supplementary Figure 7e–f, Supplementary Data 2). In contrast, chromatin-state transitions between the cell types into H3K27me3-containing states occurred at a higher frequency than expected. Notably, transition from K4 in NPC to K4/K27 in mDA (n = 236, 12.9%), occurred 2.4 times more often than expected (p = 1E-35, odds ratio, and Fisher's exact test corrected for multiple

comparisons) (Fig. 3j,k) and included genes involved in extracellular matrix organization and mitotic cell cycle (Fig. 3l, Supplementary Data 2) such as *Jag1* and *Cdk1*, respectively (Supplementary Figure 7c). In addition, K4/K27 to K4/K27 (n = 96, 5.2%) occurred 1.5 times more often than expected (p = 3E-4, odds ratio, and Fisher's exact test corrected for multiple comparisons) (Fig. 3j,m) and included genes involved in axon guidance and angiogenesis (Fig. 3n, Supplementary Data 2), such as *Reln* and *Otx2* (Supplementary Figure 7d).

Strikingly, among the most overrepresented chromatin-state transitions for genes that were downregulated from NPC to mDA were those involving the simultaneous accumulation of H3K9me3 and H3K27me3. These included transitions from K4 to K4/K9/K27 (2.9:1, p = 5E-9, n = 41, 2.2%), from K4 to K9/K27 (3.5:1, p = 0.002, n = 10, 0.54%), and also from K4/K27 to K4/K9/K27 (2.4:1, p = 7E-7, n = 44, 2.4%) and K4/K27 to K9/K27 (3.4:1, p = 2E-9, n = 34, 1.9%) (odds ratios with Fisher's exact test adjusted for multiple comparisons) (Fig. 3j). These genes were largely encoding for transcription factors important for neuronal subtype specification, neuron differentiation, and regionalization of the brain, e.g., *Zic3*, *Neurod1*, and *Sox2* (Supplementary Figure 7g,h, Supplementary Data 2), implying a layer of epigenetic control to permanently silence the fundamental NPC characteristics such as self-renewal capacity and the neurogenic machinery.

**Expansion of H3K27me3/H3K9me3 on silent genes in mDA neurons.** Notably, no downregulated gene from NPC into mDA was classified as K9/K27 in NPCs (compared to 75 in mDA) and

very few were K4/K27/K9 (36 in NPC compared to 111 in mDA) (Fig. 3j), suggesting that chromatin states including the simultaneous presence of H3K27me3 and H3K9me3 are acquired upon induction of terminal repression. To further investigate the abundance of repressive histone modifications on terminally repressed genes, we turned to analyze the set of genes that were

already silent (log2(RPKM + 1) < 1) in NPCs and remained silent in mDA neurons. Silent genes that were H3K27me3 marked in NPCs were also H3K27me3-marked in mDA (Fig. 4a). However, there was a distinct increase in H3K9me3 around broadly H3K27me3-enriched promoter regions of these silent genes (Fig. 4a, arrow), similar to what we observed for broadly

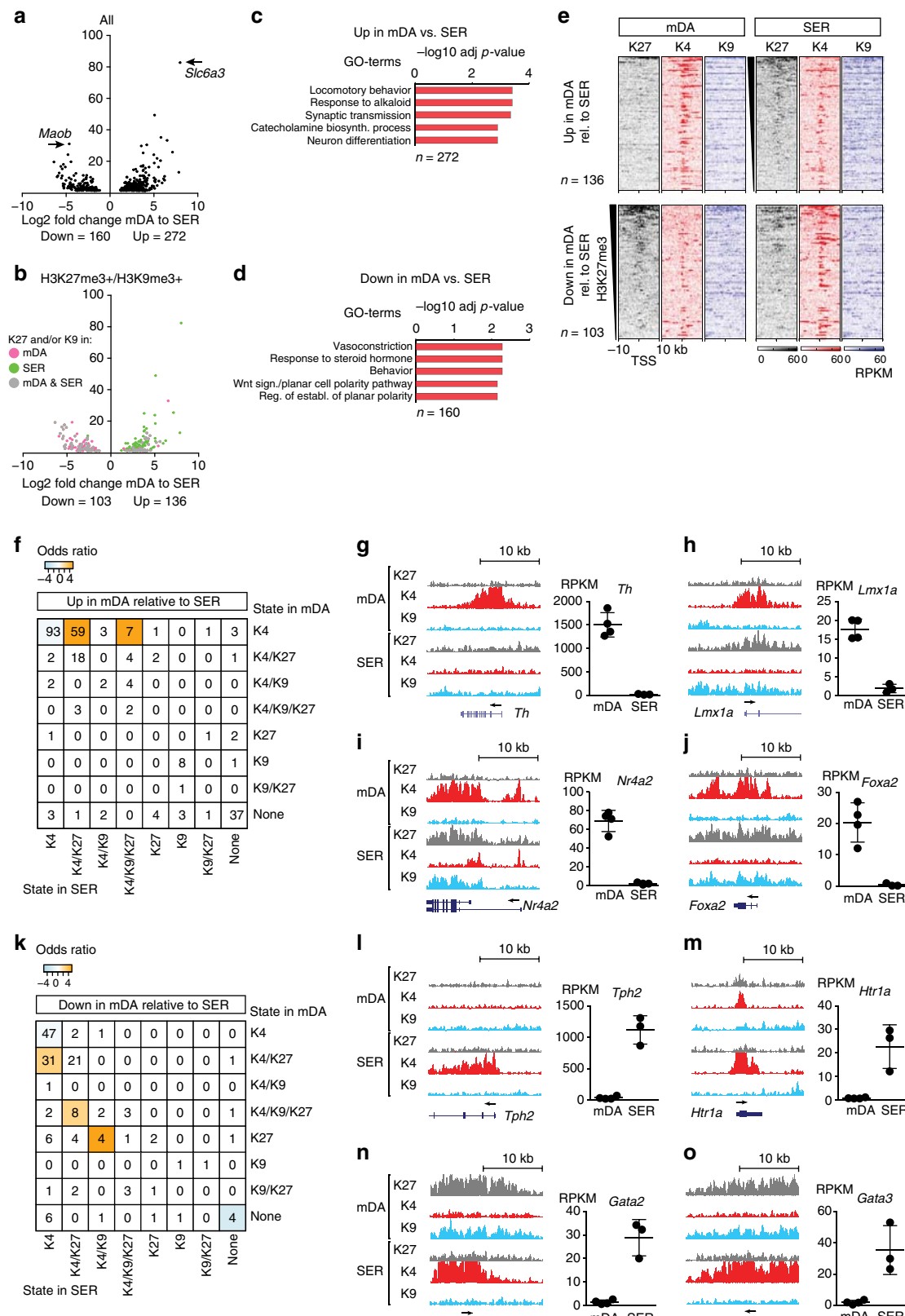

H3K27me3-enriched and downregulated genes from NPCs to mDA (Fig. 3d, arrow). Quantification of chromatin-state transitions between NPCs and mDA neurons on silent genes revealed a significant and general increase in H3K27me3- and H3K9me3-containing states in mDA relative to NPCs, as well as a significant underrepresentation in the maintenance of more active, H3K4me3-containing chromatin states (Supplementary Figure 7i). Accordingly, continuously silent genes classified as K9/K27 or K4/K9/K27 in mDA neurons showed increased enrichment of both H3K27me3 and H3K9me3 around TSS in mDA compared to NPC (Fig. 4b). This group of genes was predominantly encoding transcriptional regulators involved in early anterior/posterior pattern specification, regionalization, and embryonic morphogenesis, for example, genes within the *Hoxa*-locus (Fig. 4c,d, Supplementary Data 3). Thus, silencing of genes repressed at an earlier developmental stage than NPC is reinforced by increased deposition of H3K9me3 as cells acquire their differentiated identity.

**Subtype-specific chromatin landscapes define adult neurons.** It has been assumed that the maintenance of neuronal subtype identity is obtained by a combinatory action of specific transcription factors and cell-type-specific distribution of facultative heterochromatin[2]. Yet, the extent of H3K27me3 and H3K9me3 contribution to cell fate restriction is largely unknown[10]. To elucidate how H3K27me3 and H3K9me3 enrichment correlates with repression of alternative neuronal subtype-specific genes, we first compared gene expression differences between mDA and SER neurons. A total number of 432 genes were differentially expressed, of which 272 were upregulated in mDA neurons (mDA genes) relative to SER neurons and 160 were downregulated in mDA neurons relative to SER neurons (SER genes) (Fig. 5a). GO analysis showed that mDA genes (e.g., *Slc6a3*, *Th*, and *Ret*) were involved in the control of locomotory behavior, synaptic transmission, and catecholamine biosynthetic processes (Fig. 5c, Supplementary Data 4). Whereas, SER genes (e.g., *Tph2*, *Gata3*, and *Slc6a4*) were involved in vasoconstriction, responses to steroid hormones, and behavior (Fig. 5d, Supplementary Data 4). ChIP-seq analysis for H3K27me3, H3K4me3, and H3K9me3 on sorted SER nuclei showed that the abundance of H3K27me3 and H3K9me3 ±10 kb around TSS of genes in SER neurons was inversely correlated with abundance of H3K4me3 (Supplementary Figure 8a). Every promoter region was assigned a SER-specific chromatin state (Supplementary Figure 8b). This revealed that the expression levels of genes in different chromatin states correlated with a combinatorial presence of active and repressive histone modifications similar as for mDA neurons and NPCs (Supplementary Figure 8b–e). A substantial proportion of differentially expressed genes between SER and mDA belonged to a repressive chromatin state in either cell type (Fig. 5b).

We then examined ChIP-enrichment in regions ±10 kb around TSS of differentially regulated genes belonging to H3K9me3- and/or H3K27me3-containing states. There was a higher abundance of H3K27me3 and H3K9me3 in promoter regions of mDA genes in SER neurons and reduced H3K4me3 (Fig. 5e, upper panel). Conversely, promoter regions of SER genes were highly enriched for H3K27me3 and H3K9me3 in mDA neurons and exhibited reduced abundance of H3K4me3 (Fig. 5e, lower panel). Thus, differences in gene expression between mDA and SER neurons correlated well with cell-type-specific enrichment of H3K27me3, H3K9me3, and H3K4me3.

Quantification of differences by categorizing the differentially expressed genes between mDA and SER into chromatin states revealed that a majority of these genes fell into a state containing H3K27me3 and/or H3K9me3 (52.2% of mDA-specific genes in SER and 68.1% of SER-specific genes in mDA) (Fig. 5f,k). In fact, differentially expressed genes were less often categorized as K4 in both cell types than expected (mDA-specific genes: 0.74:1, $p = 0.001$, $n = 93$, 34.2%; SER genes: 0.64:1, $p = 2E-4$, $n = 47$, 29.4%) and much more often K4/K27 if K4 in the other cell type (mDA genes: 7.1:1, $p = 2E-31$, $n = 59$, 21.7%; SER genes: 6.0:1, $p = 5E-14$, $n = 31$, 19.4%) (odds ratios with Fisher's exact test adjusted for multiple comparisons) (Fig. 5f,k). Notably, *Th*, encoding for the rate-limiting enzyme in dopamine synthesis[29], was heavily marked by H3K4me3 in mDA neurons but lacked any of the examined repressive histone modifications in SER neurons (Fig. 5g, Supplementary Data 4). However, among the mDA-specific genes that were K4/K27 or K9/K27 in SER neurons, we identified many genes encoding for transcription factors implicated in the development and maintenance of mDA neurons such as *Lmx1a*, *Nr4a2*, and *Foxa2* (Fig. 5h–j, Supplementary Data 4)[30–33]. Correspondingly, *Tph2*, which encodes for tryptophan hydroxylase 2, the rate-limiting enzyme in the synthesis of serotonin[34] was heavily H3K4me3 marked in SER neurons but lacked any analyzed modifications in mDA neurons (Fig. 5l), while genes encoding for regulatory proteins involved in the development of SER neurons including *Gata2* and *Gata3* were K9/K27 in mDA neurons (Fig. 5n,o, Supplementary Data 4)[35–37]. Thus, genes encoding instructive transcription factors important for alternative neuronal subtype specification (e.g., *Lmx1a* and *Gata2*) exhibit a layer of repressive histone modifications, potentially to safeguard against erroneous activation. In contrast, regulation of downstream effector genes (e.g., *Th* and *Tph*) appears not to be coupled to repressive H3K27me3/H3K9me3 enrichment.

**Fig. 5** Subtype-specific chromatin landscapes define adult neurons. **a** RNA-seq: Volcano plot showing differential gene expression between SER and mDA neurons. Differentially expressed genes with an adjusted *p*-value < 0.05 obtained by Wald test are shown. **b** Volcano plot as in **a**, differentially expressed and H3K27me3 and/or H3K9me3-marked genes colored according to: marked in mDA – pink, marked in SER – green, and marked in both mDA and SER – gray. **c** Gene ontology (GO) of selected (−log10 (adj *p*-value), Fisher's exact test) biological processes for upregulated genes in mDA relative to SER (*n* = 272). **d** As in **b** for downregulated genes in mDA relative to SER (*n* = 160). **e** ChIP-seq heatmaps showing the abundance of H3K27me3, H3K4me3, and H3K9me3 as RPKMs ±10 kb around TSS of genes upregulated in mDA relative to SER neurons and associated with a chromatin state containing H3K27me3 and/or H3K9me3 in mDA and/or SER (top panels), and genes downregulated in mDA relative to SER neurons and associated with a chromatin state containing H3K27me3 and/or H3K9me3 in mDA and/or SER (bottom panels). **f** Chromatin-state transitions between SER and mDA neurons for upregulated genes in mDA relative to SER. Colored squares indicate transitions that occurred significantly more often than expected (odds ratio, orange = more than expected, blue = less than expected, and *p* < 0.05 according to Fisher's exact test and adjusted for multiple testing). **g–j** Left: UCSC Genome browser excerpts showing examples of H3K27me3- (gray, vertical scale 0–100 RPKM), H3K4me3- (red, vertical scale 0–160 RPKM), and H3K9me3 enrichment (blue, vertical scale 0–80 RPKM) around the *Th*, *Lmx1a*, *Nr4a2*, and *Foxa2* loci. Each plot spans 20 kb. Right: Bars indicate average RPKM ± SD for *Th*, *Lmx1a*, *Nr4a2*, and *Foxa2* in mDA and SER neurons, respectively. **k** Chromatin-state transitions between SER and mDA neurons for downregulated genes in mDA relative to SER as in **f**. **l–o** Left: UCSC Genome browser excerpts showing examples of enrichment of H3K27me3 (gray, vertical scale 0–100 RPKM), H3K4me3 (red, vertical scale 0–160 RPKM), and H3K9me3 (blue, vertical scale 0–80 RPKM) around the *Tph2*, *Htr1a*, *Gata2*, and *Gata3* loci. Each plot spans 20 kb. Right: Bars indicate average RPKM ± SD for *Tph2*, *Htr1a*, *Gata2*, and *Gata3* in mDA and SER neurons, respectively

A relatively large proportion of subtype-specific genes belonged to the K4/K27 bivalent-like state in the cell type where the gene was repressed (mDA genes in SER: $n = 81$, 31.5% of tot; SER genes in mDA: $n = 53$, 33.1% of tot) (Fig. 5f,k). Bivalent promoters have previously been implicated in association with spatiotemporal gene activity during development, where either the H3K4me3 or the H3K27me3 mark is lost in a cell-type-

specific manner as differentiation progresses[38,39]. Moreover, it has been shown that in the formation of adult tissues, both tissue-selective erasure and de novo acquisition of bivalency occur[40]. Our comparison between NPC and mDA showed that the K4/K27 state was both resolved and acquired during terminal differentiation of mDA neurons (Fig. 3e,j), raising the question if the K4/K27 state is cell-type specifically resolved and/or acquired

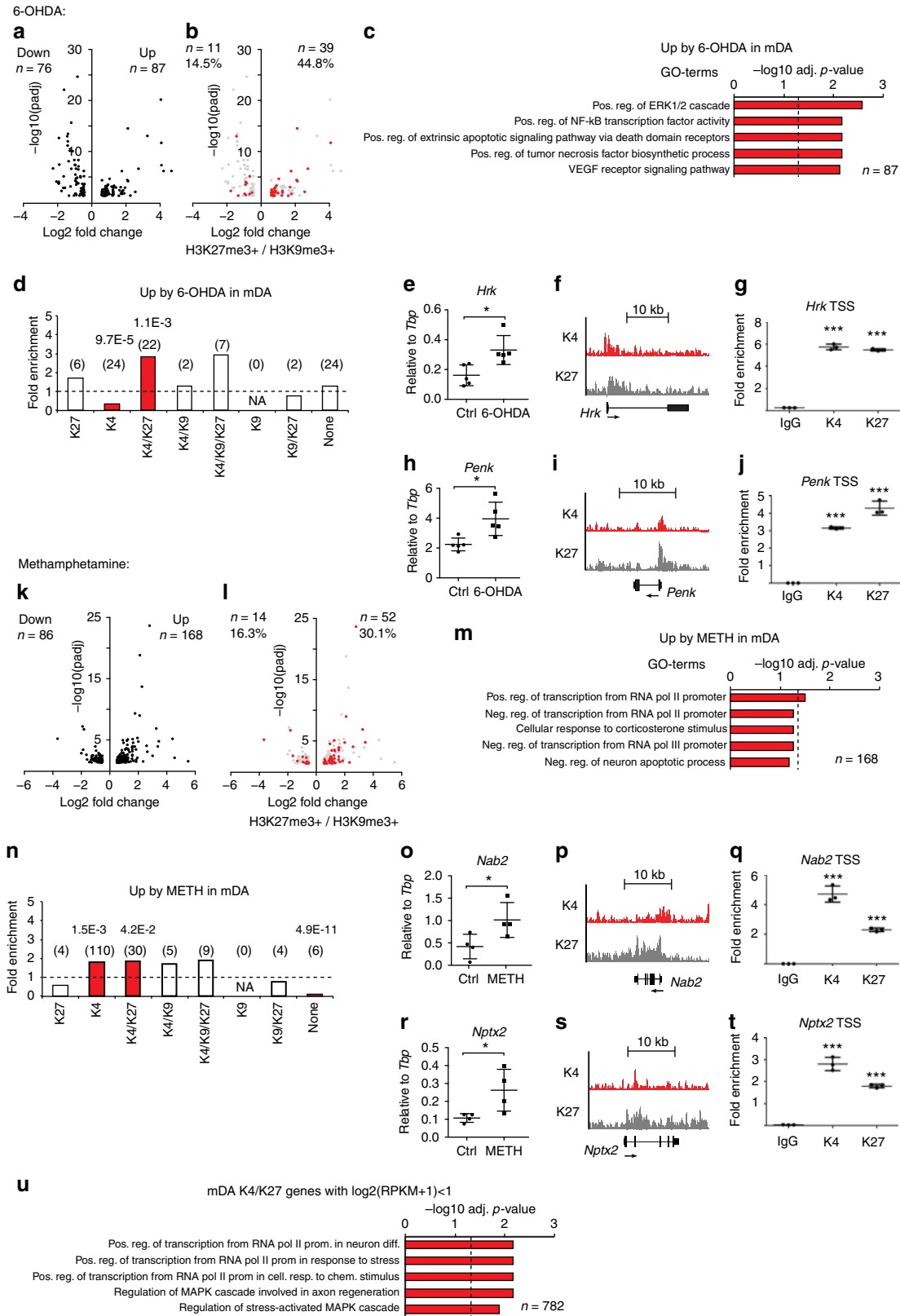

from NPC for differentially expressed genes between SER and mDA neurons. To explore this, we first analyzed the group of differentially expressed genes between SER and mDA neurons that belonged to the K4/K27 state in NPC. Genes upregulated in mDA relative to SER and K4/K27 in NPC ($n = 70$), lost H3K27me3 enrichment around TSS in mDA neurons and gained K4 to a large extent, whereas enrichment of H3K27me3 remained or increased in SER neurons (Supplementary Figure 9a, upper panel). Conversely, genes downregulated in mDA relative to SER and K4/K27 in NPC ($n = 50$) remained H3K27me3 enriched in mDA but lost H3K27me3 and gained H3K4me3 in SER (Supplementary Figure 9a, lower panel). Quantification of chromatin-state transitions between NPC and mDA for genes differentially regulated between SER and mDA showed that genes upregulated in mDA relative to SER and K4 in mDA, were 5.7 times more often K4/K27 in NPC than expected ($p = 2E-22$) (Supplementary Figure 9d). Similarly, genes downregulated in mDA relative to SER and K4/K27 in mDA were 4.3 times more often K4/K27 in NPC ($p = 5E-8$) but also 2.5 times more often K4 in NPC ($p = 0.002$) (odds ratios, Fisher's exact test adjusted for multiple comparisons) (Supplementary Figure 9e, Supplementary Data 4). This suggests that repressed genes belonging to the K4/K27 state in NPC are selectively derepressed in SER or mDA neurons and that de novo formation of the K4/K27 state occurred in both SER and mDA neurons in correlation with cell-type-specific repression.

**Activation of H3K27me3-marked genes during stress.** Many epigenetic mechanisms implicated in chromatin remodeling and developmental processes appear to retain their plasticity in the adult brain[41]. It has previously been shown that displacement of PRC2 from chromatin in neurons of the adult brain leads to aberrant activation of H3K27me3-repressed genes, either resulting from abnormal dopamine signaling through induction of H3K27me3S28 phosphorylation[13,42] or by genetic cell-type-specific ablation of the two H3K27-specific methyltransferases *Ezh1* and *Ezh2*[12]. Since repressive chromatin states were linked to the maintenance of gene expression associated with terminal differentiation as well as for neuronal subtype identity, we asked whether repressed genes associated with H3K27me3 and/or H3K9me3 could be derepressed during stressful conditions in mDA neurons. We first turned to an experimental mouse model of Parkinson's disease in which unilateral stereotaxic injection of the neurotoxin 6-OHDA results in restricted degeneration of mDA neurons, whereas the intact hemisphere serves as a within-subject control[43]. 6-OHDA lesioned *DatCreER^{T2}-Rpl10a-*

*mCherry* mice ($n = 5$) were sacrificed 14 days post surgery. Loss of TH$^{+}$-cells was confirmed in the lesioned hemisphere (Supplementary Figure 10a–c). Gene expression differences in mDA neurons between intact and lesioned hemispheres were assessed by TRAP-seq (Supplementary Figure 11a,b, Methods). We identified 163 genes that significantly changed gene expression in mDA neurons due to the 6-OHDA treatment, of which 87 genes were upregulated and 76 genes were downregulated, many of which were mDA-neuron-specific factors (e.g., *Pitx3*, *Slc6a3*, *Th*, and *En1*) (Fig. 6a, Supplementary Data 5). We then analyzed which chromatin states the regulated genes belonged to in naive mice, by correlating the TRAP-seq data to the previously obtained ChIP-seq data from mDA neurons. Notably, H3K27me3 and/or H3K9me3 marked a significant proportion of the regulated genes in mDA neurons (44.8% of upregulated genes, 14.5% of down-regulated genes) (Fig. 6b). Among other GO terms, upregulated genes enriched for positive regulation of extrinsic apoptosis (Fig. 6c). Separation of genes into their respective chromatin states in mDA neurons of naive mice, revealed that upregulated genes 2.9-times more often belonged to the K4/K27 state ($p = 1.1E-3$) (odds ratio and Fisher's exact test corrected for multiple comparisons). Other chromatin states were either under-represented or near expected ratios (Fig. 6d). To further validate that K4/K27 marked genes were induced upon 6-OHDA treatment, we confirmed this chromatin state near TSS of upregulated genes (*Hrk* and *Penk*) in mDA nuclei by ChIP-qPCR for H3K4me3 and H3K27me3 (Fig. 6e–j).

Having concluded that genes belonging to the K4/K27 state in mDA neurons were frequently induced during a neurodegenerative condition, we examined if genes belonging to repressive chromatin states were affected by other means of stress in mDA neurons. Methamphetamine is a widely used recreational drug that produces a neurotoxic effect in mDA neurons[44–46]. *DatCreER^{T2}-Rpl10a-mCherry* mice were injected with a single high dose (30 mg/kg) of methamphetamine or saline and sacrificed 2 h after injection. Gene expression changes were then assessed by TRAP-seq in the same way as for 6-OHDA (Supplementary Figure 11c,d). We found that 168 genes were upregulated and 86 genes were downregulated (Fig. 6k, Supplementary Data 5, Methods). A significant proportion of the regulated genes were marked by H3K27me3 and/or H3K9me3 in mDA neurons (30.1% of upregulated genes, 16.3% of down-regulated genes) (Fig. 6l). Among the genes upregulated by methamphetamine, many were involved in transcriptional regulation (Fig. 6m). Upregulated genes by methamphetamine were 1.8-times more often belonging to the K4 state ($p = 1.5E-3$) and 1.8-times more often belonging to the K4/K27 state ($p =$

**Fig. 6** Activation of H3K27me3 marked genes during stress. **a** TRAP-seq: Volcano plot showing differential gene expression in mDA neurons from lesioned and unlesioned hemispheres 14 days after unilateral 6-OHDA injections ($n = 5$ mice) (Methods). **b** As in **a** where H3K27me3 and/or H3K9me3-marked and regulated genes are marked with red. **c** Gene ontology (GO) of selected ($-\log10$(adj $p$-value), Fisher's exact test) biological processes for upregulated genes in mDA neurons of the lesioned hemisphere ($n = 87$). **d** Bars indicate the odds-ratio fold enrichment for upregulated genes after 6-OHDA treatments per chromatin state. Absolute numbers of genes per state are given in parenthesis. Red bars indicate significant enrichment according to Fisher's exact test and are adjusted for multiple testing. **e** TRAP-qPCR confirming induction of *Hrk* relative to *Tbp* in mDA after 6-OHDA. Bars indicate average expression ± SD. Two-tailed Student's $t$-test assuming equal variances. *$p \leq 0.05$. **f** Genome browser excerpt showing H3K4me3 (K4, red, vertical scale 0–160 RPKM) and H3K27me3 (K27, gray, vertical scale 0–100 RPKM) enrichment at the *Hrk* locus in naive mDA nuclei. **g** ChIP-qPCR in mDA nuclei showing fold-enrichment of IgG, K4, and K27 relative to input near the *Hrk* TSS. Bars indicate average enrichment ± SD. K4 enrichment was significantly higher near *Hrk* compared to near the *Hoxa2* TSS and K27 enrichment was significantly higher compared to near the *Tbp* TSS (Supplementary Fig. 11f,h). Two-tailed Student's $t$-test assuming equal variances. ***$p \leq 0.001$. **h–j** As in **e–g** for *Penk*. **k** TRAP-seq: Volcano plot showing differential gene expression in mDA neurons 2 h after a single methamphetamine injection compared to saline-injected controls, similar to **a**. **l** As in **k** where H3K27me3 and/or H3K9me3-marked and regulated genes are marked with red. **m** GO for selected ($-\log10$ (adj $p$-value), Fisher's exact test) biological processes for upregulated genes in mDA neurons after methamphetamine ($n = 52$). **n** Odds ratio for upregulated genes after methamphetamine per chromatin state as in **d**. **o–t** Examples of upregulated K4/K27 genes after methamphetamine, similar to **e–g**. **u** Silent genes with K4/K27-marked promoters are commonly involved in stress responses: GO for selected ($-\log10$ (adj $p$-value), Fisher's exact test) biological processes for K4/K27 genes with log2(RPKM + 1)<1 in naive mDA neurons

4.2E-2), whereas genes belonging to other chromatin states were either underrepresented or as expected (odds ratios and Fisher's exact test corrected for multiple comparisons) (Fig. 6n). We validated the upregulated genes *Nab2* and *Nptx2* as belonging to the K4/K27 state by ChIP-qPCR (Fig. 6o–t). In contrast, SER neurons exhibited no enriched upregulation of K4/K27 marked genes upon methamphetamine stimulation. Instead, there was a strong enrichment of K4 marked genes (Supplementary Fig. 12a–f). Thus, both the transcriptional response to neurodegenerative stress (6-OHDA) and to methamphetamine in mDA neurons is characterized by the induction of genes belonging to the K4/K27 state.

In any given cell population analyzed by ChIP-seq, an observed co-occurrence of permissive and repressive histone marks at promoters may reflect a transcriptional heterogeneity within the population. To estimate a minimal proportion of genes with true bivalence with respect to H3K4me3 and H3K27me3 around TSS in naive mDA neurons, we narrowed down the list of genes that were classified as K4/K27 based on the gene expression data and found that 31.3% (781/2497) of K4/K27 genes in mDA neurons were transcriptionally silent (log2(RPKM + 1)<1) and thus not differentially expressed within this population (Supplementary Data 6). GO-term analysis on the silent K4/K27 genes showed that these were enriched not only for developmental transcriptional regulators, but also for genes involved in stress response (Fig. 6u).

## Discussion

Previous technical limitations have restricted our knowledge about chromatin contribution to terminal gene repression in neurons to findings in cell culture. Here, we present a comprehensive map of permissive and repressive chromatin coupled to global gene expression levels in mDA neurons, NPCs, and SER neurons in vivo. Through comparative bioinformatics analysis, our study furthermore sheds light on how cell-type-specific distribution of H3K27me3, H3K4me3, and H3K9me3 provides a repressive chromatin state that potentially serves to reduce erroneous activation of alternative cell-type determinants.

We found that the different defined chromatin states were predictive of gene expression levels, as genes with promoter regions belonging to H3K4me3-containing states were generally higher expressed than H3K27me3-containing or H3K9me3-containing states, and in that order. Analyzed marks were differently distributed within the analyzed promoter regions, depending on their chromatin state. H3K27me3 was, for instance, more centered at TSS when co-occurring with H3K4me3 in the K4/K27 state, whereas H3K27me3 was covering the entire region when co-occurring with H3K9me3 in the K9/K27 state, and that these differences were reflected in cell-type transitions, gene identity, and gene-expression levels (Fig. 2b, Supplementary Figures 6b and 8b).

The majority of genes differentially expressed between NPC and mDA and between SER and mDA belonged to the K4 state in both cell types, indicating that transcription factors combined with additional chromatin modifications (e.g., DNA methylation) have a major impact on gene expression changes as a whole (Figs. 3 and 5). Still, our data support a contributing role for the examined repressive histone modifications. Among the mDA-specific genes relative to SER neurons, we found several transcription factors that are implicated in the maintenance of mDA identity and survival to various extents, including *Nr4a2 Lmx1a*, *Otx2,* and *Foxa2*[37,47–49], all of which belonged to a more repressive H3K27me3- and/or H3K9me3-containing chromatin state in SER neurons compared to mDA neurons (Supplementary Data 4). Strikingly, several genes that are governing the central

functions of mDA neurons, such as *Th* and *Ret*, were found to lack repressive marks in SER neurons despite not being expressed. However, both these genes have been shown to be regulated by *Nr4a2* in mDA neurons[37,47]. This supports a model in which cell-type-specific transcription factors continuously act to maintain the expression of cell-type-specific genes. Accordingly, transcription factors that induce and maintain expression of genes governing alternative cell types need to be repressed in a reliable manner so that aberrant expression does not activate erroneous gene programs. Genes that are repressed earlier in development, such as *Hox* genes that are already silenced in NPCs, show abundant deposition of H3K27me3 but lower levels of H3K9me3 in NPCs. However, in mDA neurons, the repressed state is reinforced by increased H3K9me3 deposition. Paradoxically, cell-cycle regulators were heavily enriched among silenced genes that did not gain any repressive histone modification yet retained the permissive H3K4me3 mark (i.e., K4 > K4). This would fit with observations of how neurons under stress can upregulate the expression of genes encoding for cell-cycle regulators[50].

Overall, the K4/K27 state was the most commonly occurring H3K27me3-containing state in the examined cell types. Our data suggest that the K4/K27 state in NPC was resolved into K4 when the gene was activated in either mDA or SER neurons (or both), and that there was a cell-type-specific acquisition of K4/K27 states on genes that were repressed from NPCs to mDA neurons, or from NPCs to SER neurons, respectively (Supplementary Figure 9). Recent studies have suggested that the main function of PRC2 in adult neurons (MSNs) and other adult tissues is to suppress transcription of genes with bivalent (H3K4me3- and H3K27me3-containing) promoters, while repression of the majority of H3K27me3-marked genes does not depend on PRC2 activity[12,40]. Our data show that predominantly H3K27me3-marked genes belonging to the K4/K27 state were activated in mDA neurons during stress, whereas induction of genes belonging to states including H3K9me3 occurred less frequently (Fig. 6). Consistent with these findings, it is therefore plausible that H3K9me3 enrichment serves as an additional layer of transcriptional repression on H3K27me3-marked genes, which could then explain that activation of primarily K4/K27-genes occurs upon ablation of PRC2[12]. This hypothesis was further supported by our observation that broadly H3K27me3-marked and silent genes in NPCs, primarily encoding early developmental transcriptional regulators, had a tendency to gain H3K9me3 during terminal differentiation into mDA neurons (Figs. 3d and 4, Supplementary Figure 7i). Although a significant subset of promoter regions belonging to the K4/K27 state in NPCs, mDA, and SER neurons likely correspond to true promoter bivalency, future studies including conditional ablation of PRC2-components and sequential ChIP experiments, may clarify the extent and contribution of truly bivalent promoters to stress response and maintenance of cell-type-specific gene expression. Furthermore, such an effort would reveal if PRC2-associated gene silencing indeed is required for maintenance of cellular identity in these cell types.

## Methods

**Ethical considerations.** All animal experiments were performed according to Swedish guidelines and regulations, and the ethical permit N189/15 was granted by "Stockholms Norra djurförsöksetiska nämnd, Sweden."

**Mice.** The generation of *DatCreER^T2^*, *SERTcre*, and *Rpl10a-mCherry* mice has been previously described[21–23]. Crosses between these mouse lines generated the *Dat-CreER^T2^-Rpl10a-mCherry*, and *SertCre-mCherry* lines used in this study. Similar numbers of males and females were used for the different analyses. Mice were kept in rooms with controlled 12-h light/dark cycles, temperature, and humidity with food and water provided ad libitum. The mice were housed at a maximum of four males per cage and six females per cage.

**Tamoxifen administration**. To induce *Rpl10a-mCherry* expression conditionally in dopamine-producing neurons, 1-month-old *DatCreER^T2^-Rpl10a-mCherry* mice heterozygous for both the floxed *Rpl10a-mCherry* alleles and the *DatCreER^T2^* transgene alleles were administered with tamoxifen (Sigma, T-5648) at 43.5 mg/kg dissolved in a 9:1 corn oil and EtOH mixture by oral gavage one dose per day for 3 consecutive days.

**Methamphetamine administrations**. Mice were injected intraperitoneally in a volume of 10 ml per kilogram of body weight with either saline or 30 mg/kg (+)–methamphetamine hydrochloride (Sigma, M8750) dissolved in saline, and were sacrificed by $CO_2$ 2 h after injections.

**6-OHDA lesions**. Mice were anesthetized in a separate cage with air supplemented with 4 mg/ml isoflurane and mounted in a stereotaxic frame (David Kopf Instruments, Tujunga, CA) equipped with a mouse adapter. Mice were kept anesthetized throughout the procedure by inhalational administration of 2 mg/ml isoflurane. 6-OHDA–HCl (Sigma-Aldrich, Sweden, AB) was dissolved in 0.02% ascorbic acid in saline at a concentration of 3 μg of free-base 6-OHDA per μl. Each mouse received one unilateral (right hemisphere) injection of 6-OHDA of 1 μl (0.5 μl/min) into the medial forebrain bundle according to the following coordinates (mm): anteroposterior (AP), −1.2; mediolateral (ML), −1.2; and dorsoventral (DV), −4.8 (all coordinates relative to bregma)[51]. Animals were sacrificed 2 weeks post surgery by $CO_2$.

**Histological analyses**. Animals were deeply anesthetized with Avertin intraperitoneal sodium pentobarbital (Apoteksbolaget, AB) and perfused with room-temperature saline (0.9% w/v) through the ascending aorta, followed by ice-cold 4% paraformaldehyde. The brains were subsequently removed, postfixed in the same fixative, and cryoprotected for 24–48 h in 25% sucrose at 4 °C before being cut on a Leica microtome at 30-μm thickness. Sections were permeabilized in 5% BSA in PBS-T (PBS with 0.2% Tween-20), followed by primary antibody incubation in a cold room overnight with sheep anti-TH (1:1.000, cat# P60101–150, Pel-Freeze) or anti-TPH2 (1:500, cat# T0678, Sigma) with an Alexa-tagged secondary antibody from Molecular Probes (1:500; donkey anti-sheep (cat# A11015) or anti-mouse (cat# A21202) IgG) for labeling midbrain dopamine-producing neurons or raphe serotonergic neurons. Section images were obtained in a confocal microscope (Axio Imager M1; Zeiss, Germany).

**FACS sorting of cell-type-specific nuclei**. *DATcreER^T2^-Rpl10a-mCherry* and *SERTcre-Rpl10a-mCherry* mice were sacrificed with $CO_2$ and the brains were quickly removed and put in PBS on ice. The midbrain or hindbrain was dissected out and snap-frozen on dry ice. A timed-pregnant (embryonic day 11.5 (E11.5)) mouse was sacrificed with $CO_2$, the embryos were removed, and sacrificed by decapitation. E11.5 brains were dissected out and snap-frozen on dry ice. Tissue was thawed and dissociated using a 1-ml dounce homogenizer (Wheaton) in ice-cold lysis buffer (0.32 M sucrose, 5 mM $CaCl_2$, 3 mM MgAc, 0.1 mM $Na_2EDTA$, 10 mM Tris-HCl, pH 8.0, 1 mM DTT, and 1x complete proteinase inhibitor, EDTA-free (Roche)). The homogenate was layered over a sucrose cushion (1.8 M sucrose, 3 mM MgAc, 10 mM Tris-HCl, pH 8.0, and 1 mM DTT) before centrifugation for 2.2 h at $30,000 \times g$ in a Beckman J-25 centrifuge using a J13.1 rotor. The pelleted nuclei were resuspended in a nuclear storage buffer (15% sucrose, 10 mM Tris_HCl, pH 7.2, 70 mM KCl, and 2 mM $MgCl_2$) supplemented with RNAse inhibitor (RNAse out, Invitrogen) and proteinase inhibitor (Complete, Roche). Resuspended nuclei were passed through a 30-μm cup filcon (BD Biosciences, 340625) into a BSA-coated tube for sorting. Nuclei from E11.5 mouse embryos were incubated with an Alexa 647-labeled antibody against SOX2 (Santa Cruz, sc-17320) using Alexa Fluor 647 Antibody Labeling Kit (Molecular Probes, P26005) overnight prior to sorting.

FACS was performed using a FACSAria cell sorter and the FACSDiva software (BD Biosciences). The nuclei were identified by forward- and side-scatter gating, and a 633-nm laser with a 660/20 or 616/23 filter. A 100-μm nozzle, sheath pressure of 20–25 psi, and an acquisition rate of up to 1000 events per second were used. The sorting strategy was confirmed to separate single nuclei from duplets/triplets: Purified nuclei from adult mouse cortex were incubated with an Alexa 647-labeled antibody against the pan-neuronal marker NeuN (Millipore, MAB377), and sorted in large numbers according to our gating strategy with the addition of DAPI to quantify DNA content per event (Supplementary Figure 13). This analysis showed that 99.5% of nuclei sorted by this method were singlets. Cell-type-specific nuclei were collected in batches of 1000, supplemented with a nuclear buffer (10 mM Tris (pH 8.0), 100 mM NaCl, 2 mM $MgCl_2$, 0.3 M sucrose, and 0.25% IGEPAL CA-630) to a total volume of 10 μl and subsequently snap-frozen on dry ice for later use.

**Bulk RNA-seq**. Total RNA was extracted from bulks of 1000 FACS-sorted nuclei using RNeasy Micro Kit (Qiagen, 74004). RNA was quantified and quality-checked with a Bioanalyzer device (Agilent) using RNA 6000 Pico Kit (Agilent, 5067–1513). Sequencing libraries were generated from 1 ng of the total nuclear RNA using the previously described Smart-seq2 protocol[52] with minor modifications, and a Nextera XT DNA sample Preparation kit (FC-131–1096) with Nextera XT indexes

(FC-131-1002). Sequencing was performed on an Illumina HiSeq2000 sequencer within the National Genomics Infrastructure at Scilife lab, Stockholm, Sweden. Raw 50-bp single-end reads were aligned to the mouse genome (mm10 assembly) using STAR v2.3.0 with default settings[53]. Library and mapping quality was assessed by analysis of cumulative gene assignment diversity and gene-body coverage (Supplementary Figure 14) using the QoRTs package[54]. Gene expression was calculated using RPKM for genes[55]. Differential gene expression was computed using DESeq2[27] with *p* adj < 0.05 and lfcThreshold = 0.5. GO terms were obtained from Enrichr (http://amp.pharm.mssm.edu/Enrichr/)[56,57]. All examples of gene expression differences in individual genes given were validated by qPCR, using the ddCt method with *Tbp* as a housekeeping gene (Supplementary Figure 15). qPCR was performed on a StepOnePlus Real-Time PCR System (Applied Biosystems) using KAPA Sybr FAST Universal qPCR kit (KAPA, KK4602). The following primers were used for this purpose: *Slc6a3_f* AACCTG-TACTGGCGGCTATG, *Slc6a3_r* GCTGACCACGACCACATAC, *Th_f* ATTGCCTTCCAATACAAGCAG, *Th_r* TAGCATAGAGGCCCTTCAGC, *Tph2_f* GAGCTTGATGCCGACCAT, *Tph2_r* GGTGAAAGCACTTA-GACTATTCCAG, *Slc6a4_f* ACCTGGACACTCCATTCCAC, *Slc6a4_r* CCTGGAGTCCCTTTGACTGA, *Nr4a2_f* TTGAATTCTCCTCCAACTTGC, *Nr4a2_r* TGAGCCCGTGTCTCTCTGT, *Nes_f* TCCCTTAGTCTG-GAAGTGGCTA, *Nes_r* GGTGTCTGCAAGCGAGAGTT, *Gata3_f* CCTATGTGCCCGAGTACAGC, *Gata3_r* ACACACTCCCTGCCTTCTGT, *Gata2_f* CACAAGATGAATGGACAGAACC, *Gata2_r* ACAGGTGCCCGCTCTTCT, *Foxa2_f* TGTGGCCCATCTATTTAGGG, *Foxa2_r* GAGCAGCAACATCACCCACAG, *Otx2_f* GGTATGGACTTGCTGCATCC, *Otx2_r* CTCTCCCTTCGCTGTTTCC, *Ccnb1_f* TCGAATCGGGGAACCTCT, *Ccnb1_r* TGCGTTAATTTTCGTGTTCCT, *Hoxa2_f* AGACCTCGACGCTTT-CACAC, *Hoxa2_r* TGGGAATGGTCTGCTCAAA, *Zic3_f* CCTGCGCAAACA-CATGAA, *Zic3_r* CTATAGCGGGTGGAGTGGAA, *Htr1a_f* CCCCTTCAGCTGTATCTTTCC, *Htr1a_r* AAAATGCAGCACGGGTTTT, *Sox2_f* GGACTTCTTTTTGGGGGACT, *Sox2_r* CAGATCTATA-CATGGTCCGATTCC, *Lmx1a_f* AGTGTGCCTCCTGCAAAGA, *Lmx1a_r* AGCCCCCACATTTGACAG, *Jag1_f* ATGTTTCTGCTGAATATTCGATCTAC, *Jag1_r* GACAGGGTTCCCATCATCC, *Sv2c_f* GCATGTCTGTCAACGGATTC, *Sv2c_r* GGAGAACACAGTGGGAATGG, *Syn1_f* TCCTCCTGCTCAACAACGA, *Syn1_r* AAAGATGTTGCTGGCCTTGT, *Tbp_f* GGCGGTTTGGCTAGGTTT, and *Tbp_r* GGGTTATCTTCACACACCATGA.

**Single-nuclei RNA-seq library preparation**. Single mCherry + cell nuclei from the dissected midbrains of two tamoxifen-treated *DATcreER^T2^-Rpl10a-mCherry* mice, one male and one female, were FACS sorted into individual wells containing 2 μl of Smart-seq2 lysis buffer, and then frozen at −80 °C. The collected nuclei were then processed using Smart-seq2 to generate cDNA libraries. cDNA tagmentation was performed using Nextera XT DNA library preparation kits (FC-131-1024) using dual indexes (i5 + i7), and sequenced on Illumina HiSeq 2000, giving 50-bp reads following demultiplexing. Reads were aligned to the mm10 assembly using default settings in STAR v2.3.0. Gene expression was calculated using RPKM for genes as for bulk RNA-seq data. In total, 98 libraries were successfully generated and sequenced.

**Single-nuclei RNA-seq analysis**. The 98 generated single-nuclei RNA-seq libraries were analyzed following a workflow contained in the R package simpleSingleCell (version 1)[58]. Briefly, nuclei with log-library sizes more than three median absolute deviations (MADs) below the median log-library size were removed from analysis. We also removed nuclei where the log-transformed number of expressed genes was three MADs below the median (Supplementary Figure 16a,b). Library and mapping quality was assessed by analysis of cumulative gene assignment diversity and gene-body coverage (Supplementary Figure 14e–h) using the QoRTs package[54]. In total, 89 nuclei passed these quality controls and were included for further analysis. Next, to improve statistical interference, low-abundance genes with an average count below a filter threshold of 1 were removed, resulting in 12235 genes remaining in the analysis (Supplementary Figure 16c). Then, library size factors were computed (according to ref. [59]) with the chosen pool sizes 10, 20, 30, and 40. The obtained size factors were tightly correlated with the library sizes for all nuclei (Supplementary Figure 16d), and the counts were subsequently normalized against these. To identify highly variable genes (HVGs[60]), a mean-variance trend was fitted against the variance of each gene using a Loess curve with span set to 0.03 (Supplementary Figure 16e). The fitted value of the trend was then used as an estimate of the technical component of the variance of expression of each gene. The biological component of the variance was then calculated by subtracting the technical component from the total variance of each gene. HVGs were then defined as genes with a biological component of variance that were significantly equal or greater than 0.5 at a false-discovery rate (FDR) of 5% (Supplementary Figure 16f). To distinguish highly correlated HVGs involved in driving systematic differences between putative subpopulations from HVGs caused by random noise, we quantified correlations between HVGs by computing Spearman's rho and the significance of each correlation was determined using a permutation test. As control for the exclusion of noncorrelated HVGs and putative sex differences between the libraries, we found that the gene *Xist* which is exclusively expressed from the inactivated X-chromosome in females[61] was among the

## Table 1 Amplification steps for ChIP-DNA library generation

| 98 °C | 4:00 |
|---|---|
| 16 cycles | |
| 98 °C | 0:30 |
| 65 °C | 0:30 |
| 72 °C | 0:30 |
| 1 cycle: | |
| 72 °C | 5:00 |
| 12 °C | ∞ |

Highest resolution shell is shown in parenthesis

identified HVGs, but was not significantly correlated to any other gene or driving any clustering into subpopulations (Supplementary Figure 16f,g). To detect bursting vs. non-bursting genes, the individual RPKMs for a number of mDA-neuron-specific genes and the *mCherry* gene in single cells were plotted against the RPKMs for *Slc6a3* for that particular cell (Supplementary Figure 17a–h).

**ChIP-seq: ChIP and library generation.** ChIP and library generation for ChIP-seq were performed using the previously described ULI-NChIP protocol[28] with minor modifications. Briefly, one batch of 1000 nuclei per animal and antibody was thawed on ice and fragmented for 10 min using MNase at 21 °C and diluted in NChIP immunoprecipitation buffer (20 mM Tris-HCl, pH 8.0, 2 mM EDTA, 15 mM NaCl, 0.1% Triton X-100, 1x EDTA-free protease inhibitor cocktail, and 1 mM phenylmethanesulfonyl fluoride (PMSF, Sigma, P7626)). Chromatin was precleared with 5 μl of a 1:1 mixture of Protein A and Protein G Dynabeads (Life Technologies, 10006D and 10007D) and IPed overnight at 4 °C with 0.25 μg of anti-H3K9me3 (Active Motif, 39161), anti-H3K27me3 (Cell Signaling, 9733), or anti-H3K4me3 (Cell Signaling, 9751). Beads were washed twice with 400 μl of a buffer containing 20 mM Tris-HCl, pH 8.0, 0.1% SDS, 1% Triton X-100, 0.1% deoxycholate, 2 mM EDTA, and 150 mM NaCl and twice with a buffer containing 20 mM Tris-HCl (pH 8.0), 0.1% SDS, 1% Triton X-100, 0.1% deoxycholate, 2 mM EDTA, and 500 mM NaCl. Protein–DNA complexes were eluted in 30 μl of a ChIP elution buffer containing 100 mM NaHCO3 and 1% SDS for 2 h at 68 °C. The eluted chromatin was purified by phenol–chloroform, precipitated with EtOH, and resuspended in 10 mM Tris-HCl at pH 8.0.

To generate libraries for sequencing, ChIPed DNA was end-repaired in a mixture of dNTP (NEB, N0447L), T4 DNA polymerase (NEB, M0203L), Klenow DNA polymerase (NEB, M0210L), and T4 PNK (NEB, M0201L) in T4 DNA ligase buffer (NEB, B0202S) for 30 min at room temperature. The product was purified with 1.8x Agencourt AMPure XP beads (Beckman Colter, A63881) and then poly-A tailed in dATP (NEB, N0440S) and Klenow (3′–5′ exo-) (NEB, M0212L) in NEB buffer 2 (NEB, B7002S) at 37 °C for 30 min followed by another 1.8x AMPure bead purification step. The product was ligated to NEBnext adapters for Illumina (NEB, E7337A) with Quick DNA ligase (NEB, M2200L) in 2x Quick DNA ligation buffer at room temperature for 1–2 h, trimmed with USER enzyme (NEB, E7338A) at 37 °C for 15 min, and subsequently purified with 0.8x AMPure beads. The adapter-ligated ChIP–DNA was then PCR-amplified using 2x Phusion HF Master mix (NEB, M0531L) and NEBNext Multiplex Oligos for Illumina, Index Primers Set 1 (NEB, E7335L) with the steps described in Table 1.

The final libraries were cleaned up using 0.8x AMPure beads, quantified on a Qubit 3.0 Fluorometer (ThermoFisher) and quality-checked on a 2100 Bioanalyzer system (Agilent) using High Sensitivity DNA kit (Agilent, 5067–4626). Libraries were sequenced on an Illumina HiSeq 2000, 50 bp per read.

**ChIP-seq: Quality control.** Two separate batches of ChIP-seq libraries were sequenced, each consisting of three biological replicates per experiment. Batch 1 consisted of a total of 36 libraries (three cell types: NPC, mDA, and SER, and four IPs: input, H3K4me3, H3K27me3, and H3K9me3). Batch 2 consisted of a total of 21 libraries (two cell types: NPC and SER, three biological replicates of IPs as in batch 1 except that H3K4me3 libraries were only generated for NPC). Reads were mapped to the mm10 mouse genome using Bowtie2 v2.2.6 with default settings[62]. Duplicate reads were marked using Picard tools v1.92. Coverage of mapped ChIP-seq libraries was generated using the tool bamCoverage in deepTools v2.2 with parameters set to ignore duplicates and to extend reads to a typical fragment length of 170 bp[63]. Signal-to-background relationships were investigated using the plot-Fingerprint tool in deepTools v2.2. Data-quality measures based on a relative phantom peak corresponding to the read length were computed using the script run_spp.R in the package phantompeakqualtools v1.1[64]. General and ChIP-seq-specific quality metrics and diagnostic graphics were generated using the package ChIPQC with default parameters, to allow for the quantitative assessment of ChIP-seq quality[65]. The quality of each library was assessed in four ways: First, we used run_spp.R and observed that all libraries passed the quality criteria; 55 libraries with the quality tag very high, one library with high, and one library with medium. Furthermore, we manually inspected all cross-correlation plots to make sure that

all libraries displayed a clear peak at a predominant fragment length separate from the phantom peak at the read length. Typically, the phantom peak was very low except for the two libraries with quality tag lower than very high. Second, all libraries except three obtained good QC measures with ChIPQC. Third, manual inspection of fingerprint plots generated with deepTools revealed that all libraries but two showed enrichment from background. Finally, manual inspection of coverage plots identified five libraries that looked noisy (one replicate each of NPC H3K4me3 and NPC H3K9me3 in batch 1, and of NPC H3K4me3, NPC H3K9me3, and SER H3K27me3 in batch 2). Notably, these five libraries included all libraries that were highlighted as problematic in the fingerprint and ChIPQC analyses. We therefore removed these five libraries and proceeded with a final set of 52 libraries. To increase coverage depth for each experiment for downstream analyses, we pooled biological replicates within each batch separately. However, since we only had mDA input experiments in batch 1, we did not pool them and instead kept them as three separate biological replicates for statistical purposes. To summarize, after pooling, we are left with 21 samples. NPC: two input, two K4, two K27, and two K9 samples. mDA: three input, one K4, one K27, and one K9 sample. SER: two input, one K4, two K27, and two K9 samples.

The median fraction of duplicated reads for these samples was 37% (range 21–65). The median fraction of unmapped reads was 10% (range 6–42). The median total number of million mapped unique reads (after removal of duplicates) for these samples were 110 for inputs (range 26–142), 22 for H3K4me3 (range 8–32), 20 for H3K27me3 (range 9–38), and 72 for H3K9me3 (range 39–164). Based on estimates of fragment lengths from both ChIPQC and run_spp.R analyses, we chose to use an estimated fragment length of 170 bp for all samples in analyses.

**ChIP-seq: Identifying marked genes and TSS chromatin states.** For each histone modification and cell type, we identified marked genes by comparing the ChIP experiments to the input experiments using the csaw package in R[66]. Reads were counted into sliding windows along the genome, with reads extended to 170-bp, low-quality reads filtered out by keeping only reads with a mapping quality of at least five, and duplicate reads removed. Window widths were set to reflect broad or more narrow distribution of the investigated histone modifications: width 1000 bp and a spacing interval of 100 bp were used for H3K27me3 and H3K9me3, whereas width 150 bp and spacing 50 bp were used for H3K4me3. Read counting was restricted to chromosomes 1–19, X and Y, and reads mapping to regions in the curated blacklist of problematic regions available from the ENCODE project were removed[67]. For each histone modification and cell type separately, we removed low-abundance windows from further analyses in two steps. First, by only keeping windows with at least 20 reads across the experiments (i.e., all input and ChIP experiments). Second, by keeping only windows displaying an enrichment of reads above the global background. Here, we used the filterWindows function in csaw with the following settings: type global, binned background of size 10000 bp, and keeping windows enriched with log2 fold-change larger than two. Composition biases across pooled libraries were normalized using the TMM method on binned counts using the same bin size of 10,000 as in the filtering and the normOffsets method. Counts for windows that passed filtering were tested for differential binding using TMM normalization, a generalized linear model, and by calculating a *p*-value for differential binding between marker and input experiments for each window using the quasi-likelihood *F*-test as described[66]. We defined TSS regions as all regions of size 20,000 bp centered on a TSS in the org.Mm.eg.db package. We used the combineOverlaps function in csaw to calculate a combined *p*-value for differential binding in each TSS region based on all windows in the region. We corrected for multiple testing to obtain a false-discovery rate (FDR) using the Benjamini–Hochberg method. We defined TSS regions as marked by a histone modification in a cell type if the TSS region was differentially bound by the marker compared to the inputs with FDR < 0.05. Finally, we assigned each TSS region a chromatin state in a cell type by combining this binary binding status of H3K27me3, H3K9me3, and H3K4me3 into the eight possible states. TSS regions were mapped to gene expression data based on gene symbols.

**Evaluation of analyzing ChIP-seq libraries from single mice.** The csaw analysis was repeated keeping libraries from individual mice as separate replicates. The outcome was compared with results obtained from using pooled libraries with the same parameter settings (Supplementary Figure 18a–c). We found a good overlap between these two approaches: 96.1% overlap for H3K4me3-positive TSS, 80.9% for H3K27me3-positive TSS, and 61% for H3K9me3-positive TSS in mDA. These overlaps were also similar in NPC and SER.

**ChIP-seq: Chromatin-state transitions between experiments.** Transitions in chromatin states between cell types were investigated using the defined chromatin states for each TSS region in each cell type. With eight chromatin states, each TSS region can be assigned a unique transition out of 64 possible chromatin-state transitions between two cell types. For each pair of cell types, we counted the number of TSS regions for each transition using all genes to generate a background 8 × 8 chromatin transition matrix. Next, we investigated if specific gene sets, such as genes displaying different gene expression levels between the cell types, were enriched for specific chromatin-state transitions. We analyzed the odds-ratio enrichment for a specific transition in a gene set by comparing the number of such

transitions in the gene set compared to the number of such transitions in the background matrix based on all genes using 2 × 2 contingency tables and Fisher's exact test. *P*-values were corrected for multiple testing into an FDR using the Benjamini–Hochberg method.

**ChIP-seq: visualizations**. ChIP-seq coverage files were generated in deepTools using the bamCoverage function. Counts were transformed to RPKMs, reads were extended to 170 bp, bin size was set to 10 bp, and duplicates were ignored. ChIP-seq heatmaps were generated from the obtained coverage files using the plotHeatmap function, and average profile heatmaps were generated using the plotProfile function. Coverage files were uploaded to UCSC Genome Browser[68] via the Galaxy platform[69] and displayed with the mean-windowing function and a smoothing window of five pixels.

**ChIP-qPCR**. mDA nuclei from a *DatCreER*$^{T2}$-*Rpl10a-mCherry* mouse were FACS-sorted into batches of 1000, and ChIP was performed as described for ChIP-seq with the addition of an antibody against rabbit IgG as control (normal rabbit IgG, Merck, 12–370). The amplified ChIPed DNA was analyzed by qPCR as for RNA-seq samples using the following primers: *Hrk*_f GAGTATCTGCTCCGCCTCAT, *Hrk*_r GAGAGAGGAACACAGAGGAGGA, *Nab2*_1_f ACCCCCTTTTCCATCA GTCT, *Nab2*_1_r GAGGTCGAAAAGAGGACGTG, *Nptx2*_2_f AGCCTGAC CTCTGACCTCCT, *Nptx2*_2_r AGCATGAGATGGCAAGATGA, *Penk*_1_f TCAATGAGAGCTTGGACCCTA, *Penk*_1_r AGATAGAATAGTCCCA GGCATCA, *Hoxa2*_f TTCCACTCCACTCGTCCTAGA, *Hoxa2*_r CCCTAGC CCAAGCTTTTGA, *Tbp*_f GATCTCTCTTTGACAGCTAGAAAGG, and Tbp_r GATTTGTTGTGCTTAAGCTGTGA. The *Tbp* and *Hoxa2* loci served as controls for H3K4me3 and H3K27me3 enrichment, respectively.

**TRAP-seq**. Tamoxifen-treated *DatCreER*$^{T2}$-*Rpl10a-mCherry* mice subjected to unilateral 6-OHDA lesions (*n* = 5) or methamphetamine/saline administration (*n* = 4) or from *SERTcre-Rpl10a-mCherry* mice subjected to methamphetamine/saline administration (*n* = 4) were sacrificed by cervical dislocation. Brains were subsequently removed and instantly dissected in ice-cold PBS and snap-frozen on dry ice for later use. Then, translating ribosomes from mDA were captured from the tissue as previously described[70]. Briefly, frozen tissue was thawed in a prechilled 1-ml tissue homogenizer on ice and homogenized with 20 dounces in a lysis buffer containing 20 mM HEPES KOH (pH 7.3), 150 mM KCl, 10 mM MgCl$_2$, and 1% (vol/vol) NP-40. EDTA-free complete proteinase inhibitor cocktail (Roche, 011873580001), 0.5 mM DTT (Invitrogen, P2325), 100 µg/ml cycloheximide (Sigma, C4859), and 10 µl/ml rRNasin Ribonuclease Inhibitor (Promega, N2511) and Superase-In (Applied Biosystems, AM2694) was added before use. A post-nuclear supernatant was obtained by centrifugation at 4 °C for 10 min at 2000 × *g* and mixed with NP-40 to a final concentration of 1% and incubated for 5 min with DHPC (1,2-diheptanoyl-*sn*-glycero-3-phosphocholine) (Avanti Polar Lipids, 850306 P) to a final concentration of 30 mM. Then, a postmitochondrial supernatant was prepared by centrifugation at 4 °C for 10 min at 20,000 × *g* and subsequently incubated with prewashed anti-RFP magnetic beads (MBL, M165–11) at 4 °C for 18 h on a rotator. After incubation, the beads were separated from the unbound fraction with a magnet and washed 4x in a buffer containing the same components as the lysis buffer but with the KCl concentration increased to 350 mM and without the addition of RNAse inhibitors. To elute immunoprecipitated RNA, the washed beads were incubated with buffer RLT from a Qiagen RNeasy kit (Qiagen, 74004), supplemented with beta-mercaptoethanol. Eluted RNA from beads and the RNA from corresponding unbound fractions was purified on RNeasy Micro columns from the same kit. RNA quality was assessed on a 2100 Bioanalyzer using a RNA 6000 Pico kit (Agilent 5067–1513). Sequencing libraries were generated using the smart-seq2 protocol in a similar way as for bulk RNA-seq described earlier and sequenced to 50-bp single-end reads on a Illumina HiSeq2500 sequencer. Mapping of reads and differential gene expression analysis was performed as for bulk RNA-seq using Deseq2[27].

TRAP specificity was validated by comparing the immunoprecipitated (TRAP) fraction to the supernatant (unbound, UB) fraction using DeSeq2 with a log2-fold cutoff of 0.5. This showed that *mCherry* as well as mDA-specific genes, including *Slc6a3* and *Th* were highly enriched in the TRAP-fraction, whereas markers for other cell types including *Gfap* (astrocytes), *Hexb* (microglia), *Mbp* (oligodendrocytes), or *Slc6a4* (SER neurons) were not enriched (Supplementary Figure 11a). To identify genes that changed expression in mDA neurons, we (1) compared differentially expressed genes between the lesioned and unlesioned hemispheres in the TRAP-fractions and (2) subtracted differentially expressed genes in the corresponding UB fractions that (3) were not enriched in TRAP compared to UB (Supplementary Figure 11b). The gene expression changes were validated by qPCR as for bulk RNA-seq for four upregulated genes using the following primers: *Tbp*_f GGCGGTTTGGCTAGGTTT, *Tbp*_r GGGTTATCTTCACACACCATGA, *Penk*_f CCCAGGCGACATCAATTT, *Penk*_r GCAGGTCTCCCAGATTTTGA, *Nptx2*_f TCAAGGACCGCTTGGAGA, *Nptx2*_r GCCCAGCGTTAGACACATTT, *Nab2*_f AGCTCCCTCTCCCACAGC, *Nab2*_r CAGCTGTAGCTCACCCAGTG, *Hrk*_f GGAGAGGGGCAGCAGACTA, and *Hrk*_r GTGATGTCTTAGTGGGGCTTCT.

**Statistical significance**. Significance for comparing gene expression for individual genes between cell types was calculated from RPKMs using two-tailed Student's *t*-test assuming equal variances. Significance for comparing gene expression between groups of genes was calculated from (log2(RPKM + 1)) RPKMs using a two-sided Wilcoxon rank-sum test. Odds ratios for chromatin-state transitions were computed using Fisher's exact test and adjusted for multiple testing.

**Data availability**. The data sets generated and analyzed during the current study are available in the GEO repository with accession number GSE107656.

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

## Acknowledgements

The Johan Holmberg lab is supported by the Swedish Children Cancer Foundation, the Swedish Cancer Foundation, Knut and Alice Wallenberg Foundation, Swedish Research Council (V.R.), The Strategic Research Programme in Cancer (StratCan, SFO), and the Ludwig Institute for Cancer Research (LICR). The computations were performed on resources provided by SNIC through Uppsala Multidisciplinary Center for Advanced Computational Science (UPPMAX) under Project SNIC b2015115. Support to Thomas Perlmann was provided by The Knut and Alice Wallenberg Foundation (grant 2013.0075) and The Swedish Research Council (VR; grant 2016-02506).

## Author contributions

E.S. and J.H. designed the study. E.S., K.To., V.R., and K.Ti. performed the experiments. E.S. and M.R. performed the bioinformatics and statistical analyses. K.Ti., Å.K.B., and T.P. provided single-cell data. E.S. and J.H. wrote the manuscript with input from all authors.

## Additional information

**Competing interests:** The authors declare no competing interests.

