## [Peer Review File · Nature Communications]

Reviewers' comments:

Reviewer #1 (Remarks to the Author):

The manuscript reports a comparative comprehensive map of permissive and repressive chromatin coupled to global gene expression levels in mDA neurons, NPCs and SER neurons in vivo using ChIP-seq combined with RNA-seq. Although these resources are very useful for the field, the manuscript suffers several drawbacks, particularly in its focus on permissive/repressive chromatin transitions among H3K4me3, H3K27me3 and H3K9me3 between NPCs and adult neurons.

- 1) The quality of cDNA library for single cell type RNA-seq is a very critical factor. The authors must show the data of cumulative gene assignment diversity and gene body coverage.
- 2) The author's analysis is centered on single nuclei RNA-seq and ChIP-seq. However, the authors have used only histone methylation in their study: unfortunately, a large number of gene expression in brain are related to the level of DNA methylation too, limiting by the design of the experiments the chances to broadly chromatin transition.
- 3) In its current form, this body of work is perhaps useful as a sequencing data of single cell type (RNA-seq and ChIP-seq of mDA neurons, SER neurons and NPCs). However, the data was not available in the manuscript, which limited a more direct analysis/review of the data. Even without an analysis of the RNA-seq and ChIP-seq data, there are important issues such as chromatin state transition that need to be addressed regarding data analysis methods. Furthermore, the data is frequently over-interpreted in this manuscript, and so some of the major conclusions are not strongly supported without suitable validation.
- 4) One of the more interesting findings of this study is the potential chromatin state differences in histone methylation e.g. K27/K4, between NPCs and adult neurons. However, it is unclear if any specific module of clustering of gene expression to support the important role of K27/K4-induced chromatin state transition between NPCs and adult neurons. I could not find a description of this approach. Furthermore, if the chromatin state differences are a major focus of the manuscript, it would be much more ideal to have biological replicates for other cell type in brain and ages being compared (e.g., same mDA neurons and SER neurons, different ages).
- 5) Functional validation may be helpful to distinct different chromatin states between NPCs and adult neurons. For instance, whether does manipulation of K27 or K4 levels affect changes in specific gene expression in NPCs, mDA neurons and SER neurons.

Minor concerns:

- 1) In figure 1B, it should be helpful to show the statistical data of FACS.
- 2) The order of supplemental figure 4 and 5 should be re-arranged.
- 3) It will be better to show a few qPCR data of mDA-, SER- or NPCs-specific genes to support the conclusion.
- 4) In Figure S2, it will be better to show FACS analysis of mCherry-positive population of total extracted mouse midbrain nuclei from SER-Cre-Rpl10a-mCherry mice.
- 5) For sorting NPCs from mouse embryos, it will be better to show FACS analysis of SOX2-positive population of total extracted mouse midbrain nuclei from SER-Cre-Rpl10a-mCherry mice.
- 6) Venn diagrams of detected up- or down-regulated mRNA genes between NPCs and adult neurons are needed.

Reviewer #2 (Remarks to the Author):

The manuscript by Sodersten et al entitled "A comprehensive map coupling distribution of histone modifications with gene regulation in adult in vivo midbrain neurons" elucidates the genome-wide chromatin modifications coupled to gene expression in dopaminergic neurons in mice. The authors use a cell-type specific ChIP-seq method combined with RNA-seq to show differential histone modifications in 3 neuronal populations: adult midbrain dopaminergic neurons, raphe nuclei

serotonergic neurons and embryonic neuronal progenitors. Their results suggest that sequential deposition of the repressive histone modifications H3K27me3 and H3K9me3 occur on developmental genes in a subtype specific manner. In addition, they show that aberrant gene expression during dopaminergic stress in a mouse model of Parkinson's disease, or after methamphetamine injection, is characterized by the de-repression of genes that are marked by H3K4me3 and H3K27me3 at their promoter regions, whereas the induction of genes with promoter regions marked by any other combination of H3K27me3 and H3K9me3 occur less frequently.

The authors employ a combination of RNA sequencing, single cell sequencing, TRAP-analysis and cell-type specific ChiP sequencing to obtain genome wide chromatin modifications coupled to gene expression in dopaminergic (mDA), serotonergic (SER), and neuronal progenitor cells (NPC). First, the authors confirm that repressive histone modifications in these cells correlates with gene suppression, while H3K4me3 is overall associated with active gene expression. Next, the authors show that the transitions in gene expression from NPCs to mDA are associated with the respective gain/loss of active/repressive chromatin modifications in vivo. Moreover, the progression of NPCs into mDA neurons is associated with the progressive gain of H3K9me3 at the TSS of silenced genes enriched for developmental transcription factors that are modified solely by H3K27me3 in NPCs and become H3K9me3/H3K27me3 in mDAs. The authors next show that gene expression differences between mDA versus SER are overall associated with the corresponding activating/suppressing histone modifications, with the interesting exception of Tph2 and Th, the two rate limiting enzymes in serotonin and dopamine synthesis, that are H3K4me3 associated in the expressing cell-type but lack any of the two repressive chromatin modifications in the reciprocal cell-type. The authors also show developmental loss/gain of previously described bivalent H3K4me3/H3K27me3 genes in NPC to mDA and SER transition. Lastly, the authors use a mDA-specific TRAP-seq approach to show that pharmacologically induced (6-OHDA and methamphetamine) degeneration of dopaminergic neurons is associated with numerous changes in gene expression that are characterized by an increased induction of genes associated with H3K4me3 and H3K4me3/H3K27me3 as compared to any of the other chromatin modifications.

This is a very comprehensive study that elucidates the neuronal cell type specific histone modifications H3K9me3, H3K27me3 and H3K4me3 in neuronal progenitors and adult dopaminergic and serotonergic neurons. The manuscript is overall very well written and easy to follow. The strength of the manuscript is the elegant combination of cell-type specific RNA, single cell RNA, and chromatin analysis of the small number of dopaminergic/serotonergic neurons in mice. The major weakness of the manuscript is its descriptive nature and the lack of novelty. While the elucidation of genome-wide chromatin modifications coupled to gene expression in dopaminergic neurons is technically advanced, the overall conclusions of their findings are neither unexpected nor novel.

Major technical comments:

- Using the FACS-based nuclei isolation/sorting approach, it is important to separate single + nuclei from duplets and triplets (which could contain a + nuclei plus contaminating - nuclei) that occur frequently during this procedure. To ensure single nuclei isolation, a DNA dye (i.e. Dye-cycle ruby) is added that allows the FACS isolation/gating for singlets based on DNA content. According to the methods section, it is not clear if the authors performed this step.
- For the TRAP analysis of mDA in response to neurodegeneration, the gene lists contains a large fraction of genes associate/enriched in glia cells. The glia specific genes that include GFAP (astrocytes) and HexB (microglia) suggest changes in the surrounding cell/ types/tissue that is not specific to mDA but rather reflects the background of the IP. It is not clear if the authors did perform input/unbound subtraction of the TRAP sample, as suggested by Doyle et al. 2008, Cell. This is an important step in the TRAP protocol that should always be performed, especially if gene expression in the surrounding cells/tissue changes dramatically between the two conditions. Otherwise the analysis of genes that change expression in glia cells in comparison with mDA neuron specific chromatin modifications becomes non-interpretable.
- The authors mention the bias towards bivalent H3K27me3/H3K4me3 genes that are induced in

response to dopaminergic neuron stress. While this is an interesting observation, the proof for bivalency on the specific genes would require a sequential ChIP experiments, the number of mice required to generate these data stands in no relation with the information obtained, especially since the existence of bivalent states have been previously shown for other mature neurons. However, the authors should try to narrow the list of true bivalent genes in mDA (where the TSS of the gene is bivalently modified but genes are not expressed) and exclude genes that are likely differentially expressed within the population (some cells are H3K4me3 positive and gene is expressed while others are H3K27me3 positive with repressed gene expression) based on gene expression data from the same cells.

Minor comments:

- The authors state that they use Rpl10a-mCherry mouse line for nuclei isolation followed by ChIP-seq. The authors state vaguely that this line has partially nuclear localization and this is the reason they could sort nuclei from total ex-vivo isolated nuclei. This approach has been previously described by Kriaucionis et al, Science, 2009 and the possibility to use TRAP mice for nuclei isolation is based on the assembly of ribosomes, including the tagged ribosomal protein Rpl10a-mCherry, in the nucleoli of the cell nucleus.
- In figure 6, can the authors add a validation of their 6-OHDA experiments? How many of the dopaminergic neurons were destroyed?
- In figure 6H, It would add to the specificity of the methamphetamine injections if the authors performed TRAP on serotonin neurons to show that they affect a neurotoxic effect on mDA neurons and not a random effect of the drug injection.
- Since most of the data in this paper is correlative between histone methylation and gene expression, it would be beneficial if there would be some kind of manipulation of histone methylation and therefore a change in gene expression within the studied cell types. Does manipulating histone methylation in mDA neurons change their fate? Will they become more similar to serotonin neurons? Or other neurons?

Reviewer #3 (Remarks to the Author):

Review of Sodersten, et al., "A comprehensive map coupling distribution of histone modifications with gene regulation in adult in vivo midbrain dopamine neurons."

The major claims of the paper are to report differential deposition of repressive (H3K27me3 and H3K9me3) histone modifications in dopaminergic and serotonergic neurons, relative to NPCs, that were acquired during development. The authors combined several cutting-edge techniques to accomplish measurement of multiple histone modifications from pools of 1000 neuronal nuclei. First, they report that mCherry protein expression associated with TrapSEQ vectors also labels midbrain dopaminergic (mDA) and Raphe serotonergic (Ser) nuclei. They show good co-localization of mCherry and known cell-type markers by immunostaining, and perform single cell RNAseq of mDA nuclei to further define the specificity of this approach. Second, they then sort multiple 1000 nuclei pools from mDA, Ser, and NPCs and perform native ChIPseq and nuclear RNAseq on pools to generate H3K27me3, H3K9me3, H3K4me3 profiles and transcriptomic data for each cell type. Third, they utilize TrapSEQ to determine gene expression differences after administration of the neurotoxin 6-OHDA and methamphetamine. They then employ their map of naïve chromatin states to understand differential gene expression and identify bivalent K4me3/K9me3 marks as predictors of differentially expressed genes. Overall, this is a very well done study and it presents an important coalescence of methods toward an approach that should be broadly utilized by neuroscientists and others who study small cell populations. One major criticism and a few smaller criticisms are noted, with these addressed I recommend accepting the manuscript with revision.

Major Criticism:

1) The finding that bivalent promoters are “primed” for differential gene expression after 6-OHDA and methamphetamine is important. The author’s should confirm the chromatin state near at least a subset of differentially expressed genes to complete the study.

Minor Criticisms:

1) As an important consolidation of multiple approaches, data that permit a better understanding of potential pitfalls are important. The author’s should use mCherry expression in single cell RNAseq data to determine a) if all marker gene expressing cells also express mCherry and b) if the three outliers fail to express mCherry. This is important to identify FACS vs. reporter gene as a source of error.

2) Application of multiple ChIPseq and RNAseq assays on the small cell populations from the same individual mouse is a significant advance for neuroscience. However, it seems that the authors do not actually accomplish this. As I read the methods, one or two pools of three mice were pooled to generate profiles. The applicability of the approach to single mice should be evaluated and discussed.

3) To a broad readership, it is not made clear why highly variable genes (HVG) were used for analysis of single cell RNAseq. This should be clarified.

4) The transition-state matrices (e.g. Fig 3E, J) may be an appropriate approach, but I cannot discern how significance of certain states over others was determined. This should be clarified.

Response to Reviewers' comments:

We appreciate the possibility to submit a revised version of our manuscript to Nature Communications. We are also grateful for the positive response and constructive feedback that we received from the referees. Their insightful comments helped us to substantially strengthen the manuscript and we hope that you will find it suitable for publication in Nature Communications. Please see below for our point-by-point response to the issues raised.

Reviewer 1

Major points:

1) *The quality of cDNA library for single cell type RNA-seq is a very critical factor. The authors must show the data of cumulative gene assignment diversity and gene body coverage.*

We agree with the reviewer and have added a supplementary figure (**Supplementary Fig. 14**) where we show that the cumulative gene assignment diversity and gene body coverage appears as expected for RNA-seq libraries obtained by the Smart-seq2 protocol.

2) *The author's analysis is centered on single nuclei RNA-seq and CHIP-seq. However, the authors have used only histone methylation in their study: unfortunately, a large number of gene expression in brain are related to the level of DNA methylation too, limiting by the design of the experiments the chances to broadly chromatin transition.*

We agree with the reviewer that DNA-methylation, as well as several other modifications, also are associated with differences in levels of gene expression. This is underscored in the discussion on p21: "The majority of genes that were differentially expressed between NPC and mDA and between SER and mDA belonged to the K4-state in both cell types, indicating that transcription factors combined with additional chromatin modifications (e.g. DNA methylation) have a major impact on gene expression changes as a whole (**Figure 3 and 5**). " Our manuscript is a proof-of-concept study wherein we have focused on three histone modifications in restricted adult neuronal populations. The design of the study allows for analysis of additional modifications (e.g. DNA-methylation) and other cell populations, as is made clear in the introduction on page 5: "This strategy can be generalized for the identification and functional characterization of additional mechanisms involved in the maintenance of gene expression in other classes of neurons." However, we believe that this falls outside the scope of this particular study.

3) *In its current form, this body of work is perhaps useful as a sequencing data of single cell type (RNA-seq and CHIP-seq of mDA neurons, SER neurons and NPCs). However, the data was not available in the manuscript, which limited a more direct analysis/review of the data.*

We have submitted all the raw data (212 libraries in total plus additional processed data: 52 ChIP-seq, 10 RNA-seq, 98 single nuclei RNA-seq, 52 TRAP-seq) to GEO with accession number GSE107656 which can be accessed here: <https://www.ncbi.nlm.nih.gov/geo/query/acc.cgi?acc=GSE107656>

The following secure token has been created to allow review of record GSE107656 while it remains in private status: "qhuxsowarfzlod".

Even without an analysis of the RNA-seq and ChIP-seq data, there are important issues such as chromatin state transition that need to be addressed regarding data analysis methods.

We apologize for not being clear enough in defining the term "chromatin state". In the original manuscript, this was described in the methods section. In the results section we have now also added a reference to a study in Cell Stem Cell (Hawkins et al 2010), wherein the term is defined page 6. For a full description of the statistical methods used to determine chromatin state transition significance, please see methods, subheading: "ChIP-seq: Identifying marked genes and TSS chromatin states" on page 34 and subheading "ChIP-seq: Chromatin state transitions between experiments" on page 35.

Furthermore, the data is frequently over-interpreted in this manuscript, and so some of the major conclusions are not strongly supported without suitable validation.

To validate RNA-seq and ChIP-seq data we have performed qPCR and ChIP followed by qPCR on selected genes. These experiments are described in (Fig. 6 and Supplementary Fig. 15) and we believe that our major conclusions now have suitable validations.

4) *One of the more interesting findings of this study is the potential chromatin state differences in histone methylation e.g. K27/K4, between NPCs and adult neurons. However, it is unclear if any specific module of clustering of gene expression to support the important role of K27/K4-induced chromatin state transition between NPCs and adult neurons. I could not find a description of this approach*

We agree with the reviewer that this is an important question and we understand that we were somewhat vague when describing this in our figures. In Figs. 3, 4, 5 and Supplementary Fig. S7 we show the gene categories that are associated chromatin state transitions and up- or down-regulated in mDA vs NPCs. To underline that these categories are gene ontology (GO) categories we have now added "GO-terms" as a distinct heading in the figures. This was previously only written in the figure legend.

Furthermore, if the chromatin state differences are a major focus of the manuscript, it would be much more ideal to have biological replicates for other cell type in brain and ages being compared (e.g., same mDA neurons and SER neurons, different ages).

We agree with the reviewer that it would be interesting to investigate possible changes in chromatin states in additional cell types over time. Our study shows that that indeed would be feasible, as is highlighted in the introduction on page 5: "This strategy can be generalized for the identification and functional characterization of additional mechanisms involved in the maintenance of gene expression in other classes of neurons." However, we believe that this falls outside the scope of this particular study.

5) *Functional validation may be helpful to distinct different chromatin states between NPCs and adult neurons. For instance, whether does manipulation of K27 or K4 levels affect changes in specific gene expression in NPCs, mDA neurons and SER neurons.*

This is indeed an intriguing question, but would require a conditional mutant of obligate PRC2-members (for H3K27me3), TrxG-members (for H3K4me3) and SET-domain containing methyltransferases (for H3K9me3) crossed into the double mutants that we utilize in the current study. This would require a substantial amount of time. Furthermore, in a Nature Neuroscience article from 2016 the Schaefer group shows that deletion of PRC2 activity in differentiated medium spiny neurons requires >6 months before a phenotype is visible (von Schimmelmann et al 2016). Thus, it would probably take up to two years before conclusive data could be generated to answer this question. This we believe puts it outside of the scope of the study. We also added a paragraph discussing potential functional consequences of loss-of PRC2 activity on page 23: "Interestingly, in a recent study it was shown that upon loss of PRC2 activity in MSNs, bivalent genes were enriched among de-repressed whereas genes harboring H3K27me3 without H3K4me3 remained largely repressed [12]. In addition, even though there was an induction of non-MSN transcriptional regulators, there was no pronounced transdifferentiation into other neuronal subtypes. This implies that in MSNs, alternative gene programs are repressed by additional mechanisms and/or that the activation of non-MSN transcriptional regulators lacked the specificity to fully induce acquisition of an alternative cellular identity. It remains to be investigated if loss of PRC2-activity would have the same consequences for gene expression and phenotypic stability in mDA or SER-neurons."

Minor points:

1) *In figure 1B, it should be helpful to show the statistical data of FACS.*

We agree with the reviewer and have added the data in (**Supplementary Fig. 2**).

2) *The order of supplemental figure 4 and 5 should be re-arranged.*

We have changed the order of **S4** and **S5** as suggested by the reviewer, they now appear as **Supplementary Fig. 8** and **7**.

3) *It will be better to show a few qPCR data of mDA-, SER- or NPCs-specific genes to support the conclusion.*

We have performed qPCR as suggested by the reviewer and this is described in **Supplementary Fig. 15**.

4) *In Figure S2, it will be better to show FACS analysis of mCherry-positive population of total extracted mouse midbrain nuclei from SER-Cre-Rpl10a-mCherry mice.*

We now show FACS-plots as requested by the reviewer, see **Supplementary Fig. 5**.

5) *For sorting NPCs from mouse embryos, it will be better to show FACS analysis of SOX2-positive population of total extracted mouse midbrain nuclei from SER-Cre-Rpl10a-mCherry mice.*

We now show FACS-plots as requested by the reviewer, see **Supplementary Fig. 4**.

6) *Venn diagrams of detected up- or down-regulated mRNA genes between NPCs and adult neurons are needed.*

We now show Venn diagrams as requested by the reviewer, see (**Supplementary Fig. 9**).

Reviewer #2

Major technical comments:

- *Using the FACS-based nuclei isolation/sorting approach, it is important to separate single + nuclei from duplets and triplets (which could contain a + nuclei plus contaminating – nuclei) that occur frequently during this procedure. To ensure single nuclei isolation, a DNA dye (i.e. Dye-cycle ruby) is added that allows the FACS isolation/gating for singlets based on DNA content. According to the methods section, it is not clear if the authors performed this step.*

We agree with the reviewer and have now added an additional supplementary figure describing the procedure (**Supplementary Fig. 13**). This analysis shows that within the gate chosen for sorting 99.5% of the nuclei are singlets. Notably, even outside of this gate 92.7% are singlets implying efficient preparation of single nuclei.

- *For the TRAP analysis of mDA in response to neurodegeneration, the gene lists contains a large fraction of genes associated/enriched in glia cells. The glia specific genes that include GFAP (astrocytes) and HexB (microglia) suggest changes in the surrounding cell/ types/tissue that is not specific to mDA but rather reflects the background of the IP. It is not clear if the authors did perform input/unbound subtraction of the TRAP sample, as suggested by Doyle et al. 2008, Cell. This is an important step in the TRAP protocol that should always be performed, especially if gene expression in the surrounding cells/tissue changes dramatically between the two conditions. Otherwise the analysis of genes that change expression in glia cells in comparison with mDA neuron specific chromatin modifications becomes non-interpretable.*

We have now repeated the TRAP experiment and performed the suggested subtraction of the unbound fraction. The procedure is described in the text on page 17 and in the methods section on pages 37-38. As suggested by the reviewer, the enrichment of typical glial or oligodendroglial genes is no longer present (**Supplementary Fig. 11**). It is obvious that the new TRAP experiment generated a much smaller proportion of differentially expressed genes than the original experiment. This was however not a consequence of the suggested removal of unbound fraction as described above, since that procedure only removed roughly 25% of the enriched genes. One factor possibly affecting the lower number of detected genes is that the antibody we used in the first experiment (M165-9) was discontinued from the supplier and that the replacement antibody (M165-11) may be less efficient for IP. The quality of new data was however sufficient to reproduce the main finding in **Figure 6** that H3K27me3 genes are de-repressed during stress in mDA-neurons, and to demonstrate the usability of our ChIP-seq map for this type of analysis.

- *The authors mention the bias towards bivalent H3K27me3/H3K4me3 genes that are induced in response to dopaminergic neuron stress. While this is an interesting observation, the proof for bivalency on the specific genes would require a sequential ChIP experiments, the number of mice required to generate these data stands in no relation with the information obtained, especially since the existence of bivalent states have been previously shown for other mature neurons. However, the authors should try to narrow the list of true bivalent genes in mDA (where the TSS of the gene is bivalently modified but genes are not expressed) and exclude genes that are likely differentially expressed within the population (some cells are H3K4me3 positive and gene is expressed while others are H3K27me3 positive with repressed gene expression) based on gene expression data from the same cells.*

We agree with the reviewer and have now generated a list of silent genes that harbor both H3K4me3 and H3K27me3 around the TSS of mDA-neurons (**Supplementary table 6**). GO-analysis shows that these genes are associated with cellular responses to stress (**Figure 6u**).

Minor comments:

- *The authors state that they use Rpl10a-mCherry mouse line for nuclei isolation followed by ChIP-seq. The authors state vaguely that this line has partially nuclear localization and this is the reason they could sort nuclei from total ex-vivo isolated nuclei. This approach has been previously described by Kriaucionis et al, Science, 2009 and the possibility to use TRAP mice for nuclei isolation is based on the assembly of ribosomes, including the tagged ribosomal protein Rpl10a-mCherry, in the nucleoli of the cell nucleus.*

We have now added a reference to Kriaucionis et al Science, 2009, describing this procedure (page 6).

- *In figure 6, can the authors add a validation of their 6-OHDA experiments? How many of the dopaminergic neurons were destroyed?*

We have added a supplementary figure describing the loss of dopaminergic neurons upon 6-OHDA treatment (**Supplementary Fig. 10**).

- *In figure 6H, It would add to the specificity of the methamphetamine injections if the authors performed TRAP on serotonin neurons to show that they affect a neurotoxic effect on mDA neurons and not a random effect of the drug injection.*

We have added a supplementary figure describing the effects on the serotonergic population (**Supplementary Fig. 12**) this is described in the manuscript on p19.

- *Since most of the data in this paper is correlative between histone methylation and gene expression, it would be beneficial if there would be some kind of manipulation of histone methylation and therefore a change in gene expression within the studied cell types. Does manipulating histone methylation in mDA neurons change their fate? Will they become more similar to serotonin neurons? Or other neurons?*

This is indeed an intriguing question, but would require a conditional mutant of obligate PRC2-members (for H3K27me3), TrxG-members (for H3K4me3) and SET-domain containing methyltransferases (for H3K9me3) crossed into the double mutants that we utilize in the current study. This would require a substantial amount of time. Furthermore, in a Nature Neuroscience article from 2016 the Schaefer group shows that deletion of PRC2 activity in differentiated medium spiny neurons (MSN) requires >6 months before a phenotype is visible (von Schimmelman et al 2016). Thus, it would probably take up to two years before conclusive data could be generated to answer this question. This we believe puts it outside of the scope of the study.

As we point out in the discussion, in the Schaefer study on MSN there is no overt signs of transdifferentiation to other type of neurons. Rather, there is loss of MSN specific gene expression and an increase in expression of H3K27me3/H3K4me3 marked bivalent genes associated with neurodegeneration. If this is the case also for dopamine neurons remains to be investigated. See discussion on p23: “Interestingly, in a recent study it was shown that upon loss of PRC2 activity in MSNs, bivalent genes were enriched among de-repressed whereas genes harboring H3K27me3 without H3K4me3 remained largely repressed [12]. In addition, even though there was an induction of non-MSN transcriptional regulators, there was no pronounced transdifferentiation into other neuronal subtypes. This implies that in MSNs, alternative gene programs are repressed by additional mechanisms and/or that the activation of non-MSN transcriptional regulators lacked the specificity to fully induce acquisition of an alternative cellular identity. It remains to be investigated if loss of PRC2-activity would have the same consequences for gene expression and phenotypic stability in mDA or SER-neurons.”

Reviewer #3

Major criticism:

1) *The finding that bivalent promoters are “primed” for differential gene expression after 6-OHDA and methamphetamine is important. The author’s should confirm the chromatin state near at least a subset of differentially expressed genes to complete the study.*

We agree with the reviewer that this confirmation is important. We have sorted additional mDA nuclei and performed ChIP-qPCR on selected targets that were assigned to the K4/K27-state in mDA and regulated by either 6-OHDA or methamphetamine. The data is described in **(Figure 6e-f and m-n)**. We have also added genome browser excerpts showing ChIP enrichment around the target genes **(Figure 6g-h and o-p)** as well as TRAP-qPCR to confirm the gene expression changes.

Minor criticisms:

1) *As an important consolidation of multiple approaches, data that permit a better understanding of potential pitfalls are important. The author’s should use mCherry expression in single cell RNAseq data to determine a) if all marker gene expressing cells also express mCherry and b) if the three outliers fail to express mCherry. This is important to identify FACS vs. reporter gene as a source of error.*

We understand the reviewer’s concern regarding the specificity of the nuclei analyzed. We analyzed *mCherry* mRNA expression in our single-cell data and found that the three nuclei determined to be non-mDA nuclei do not have reads mapping to *mCherry*. However, not all of the 86 nuclei determined as mDA nuclei have reads mapping to *mCherry* despite detectable Rpl10a-mCHERRY in FACS. This is expected as single cells/nuclei analysis show that RNA transcripts are not constantly generated but exhibit a bursting behavior (Raj, A., Peskin, C. S., Tranchina, D., Vargas, D. Y. & Tyagi, S. Stochastic mRNA synthesis in mammalian cells. PLoS Biol. 4, e309 (2006). Suter, D. M. et al. Mammalian genes are transcribed with widely different bursting kinetics. Science 332, 472-474 (2011). Reinius och Sandberg NRG (2015)). In addition, the effect of bursting on the number of detected genes per individual cell is likely to be accentuated in nuclear RNA-seq compared to whole cell RNA-seq. However, we believe that our PCA analysis together with the combined expression of several mDA-specific genes (e.g. *Th*, *Slc6a3* (*Dat*), *Nr4a2*, *Foxa1*, *Foxa2*, *Lmx1a* and *Lmx1b*) is a strong argument for that the 86 nuclei represent a coherent mDA population. It should be noted that several dopaminergic genes in the mDA population, exhibit a similar bursting behavior, e.g. all nuclei that express *Th* or *Slc6a3* (*Dat*) do not necessarily produce the mDA-

obligatory factors *Foxa1*, *Foxa2*, *Lmx1a*, *Lmx1b* mRNA simultaneously at a given time-point. We have added a supplementary figure describing this, (**Supplementary Fig. 17**).

2) Application of multiple ChIPseq and RNAseq assays on the small cell populations from the same individual mouse is a significant advance for neuroscience. However, it seems that the authors do not actually accomplish this. As I read the methods, one or two pools of three mice were pooled to generate profiles. The applicability of the approach to single mice should be evaluated and discussed.

We appreciate the reviewer's insightful comment and his/her careful reading of the manuscript. In ChIP-seq, the proportion of uniquely mapped reads is indicative of genomic coverage saturation. Thus, some very broad histone marks, including H3K9me3, ideally require a much larger sequencing depth than less distributed marks, e.g. H3K4me3 (REF: <https://www.ncbi.nlm.nih.gov/pmc/articles/PMC4027199/>). In line with this, we noted that the proportion of uniquely mapped reads in individual H3K9me3 libraries was high relative to other marks and therefore reasoned that pooling would increase coverage per sample.

To show that our approach indeed can be implemented using single mice as well, we have repeated the analysis associated with H3K4me3, H3K27me3 and H3K9me3 enrichment around TSS in the different cell types. This analysis is described in (**Supplementary Fig 18 and methods page 35**). The overlap between detected genes when utilizing the original pooled strategy and when treating each mouse as an individual sample is good, especially for H3Kme4 associated genes. Taken together, we believe that our analysis show that it is feasible to utilize single mice for combining RNAseq with several ChIP-seq experiments.

3) To a broad readership, it is not made clear why highly variable genes (HVG) were used for analysis of single cell RNAseq. This should be clarified.

We have added a reference on page 6 describing the use of HVG for analysis of the single cell RNAseq.

4) The transition-state matrices (e.g. Fig 3E, J) may be an appropriate approach, but I cannot discern how significance of certain states over others was determined. This should be clarified.

For a full description of the statistical methods used to determine chromatin state transition significance, please see methods, subheading: "ChIP-seq: Chromatin state transitions between experiments" on page 35.

REVIEWERS' COMMENTS:

Reviewer #1 (Remarks to the Author):

The authors of the manuscript entitled "A comprehensive map coupling histone modifications with gene regulation in adult dopaminergic and serotonergic neurons" have carefully taken all of my suggestions into consideration. I really appreciate the thorough effort to do a lot more work that enhances the manuscript. Particularly, the performance of qPCR and ChIP followed by qPCR on selected genes. In sum, the authors have addressed my concerns and I think this manuscript is suitable for the general readership of Nature Communication.

Reviewer #2 (Remarks to the Author):

The manuscript from Soedersten et al. entitled "A comprehensive map coupling histone modifications with gene regulation in adult dopaminergic and serotonergic neurons" includes numerous additions that address several of the initial concerns about the fidelity of the experimental approaches. The authors added a detailed description and validation of the neuronal nuclei isolation. The authors repeated the TRAP experiments with the appropriate bioinformatic subtraction of the unbound fraction. This procedure greatly improved the quality of the dopaminergic neuron-specific (mDA) expression data and removed the majority of non-mDA genes likely coming from the surrounding glia cells.

The described RNA expression approaches, that include single cell RNA sequencing to validate cell purity followed by bulk RNA sequencing, have been used to establish comprehensive mDA and serotonergic (SER) neuron-specific gene expression data. To address how these mDA and SER specific gene expression patterns are correlated with specific chromatin modifications (H3K4me3, H3K9me3, and H3K27me3) during developmental transitions and in the adult, the authors performed low-input chromatin immunoprecipitation followed by genome wide sequencing. In summary, the authors found a significant correlation between the presence of H3K27me3 and H3K27me3/H3K9me3 at genes that are repressed in adult mDA and SER neurons, and these suppressive chromatin modifications are inversely correlated with H3K4me3 and active gene expression.

Next, to address if mDA genes associated with repressive chromatin modifications can be induced in response to cellular stresses, the authors perform mDA neuron-specific gene expression analysis after treatment with two different neurotoxins. The authors claim that H3K4me3+/H3K27me3+ bivalent loci in mDA are particularly "primed" to respond to cellular stressors such as 6-OHDA or methamphetamine (Figure 6). Using the mDA neuron specific TRAP approach, the authors identify 87 genes that are induced in mDA after exposure to the neurotoxic 6-OHDA injection. To identify truly bivalent K4me3/K27me3 genes in mDA -in contrast to genes that are differentially expressed within the mDA neuron population- the authors first excluded all genes that showed significant RNA expression in mDAs. Using this approach, the authors identified ~800 truly bivalent genes - among the original 2520 genes associated with both H3K4me3 and H3K27me3- that are transcriptionally silent in mDA neurons (Suppl. Table 5). Notably, this number of bivalent genes is similar to previously published data in adult neurons in the striatum.

The conclusion that neurodegeneration-inducing cellular stress, such as 6-OHDA, leads to preferential induction of bivalent genes, however, is not supported by the data shown in Fig 6d. While the authors show a significant enrichment for mDA genes associated with H3K4me3/H3K27me3 among the 87 genes upregulated by 6-OHDA treatment, only 3/22 genes (Hrk, CD14, Mafb) (Suppl. Table 6) are truly bivalent genes based on the authors classification (Suppl. Table 5). The remaining genes are transcriptionally active and likely reflect differential gene expression and associated chromatin modification within the mDA cell population. Without these data supporting the physiological importance of "primed" bivalent genes in response to stress, the manuscript, while technically elegant and greatly improved, remains descriptive.

Reviewer #3 (Remarks to the Author):

The manuscript by Sodersten, et al. is both important and well-done. They have addressed my concerns and I agree with their assertion that some other reviewer comments are beyond the scope of this study. One comment that I suggest for final revisions is that the authors recognize that while cell state in neurons appear permanent, cell type changes have been reported. Thus, I suggest a qualifier " . . largely retain their identity" be used in text. I recommend publication of the manuscript at this point.

Response to Reviewers' comments:

Response to reviewer #2: The manuscript from Soedersten et al. entitled “ A comprehensive map coupling histone modifications with gene regulation in adult dopaminergic and serotonergic neurons” includes numerous additions that address several of the initial concerns about the fidelity of the experimental approaches. The authors added a detailed description and validation of the neuronal nuclei isolation. The authors repeated the TRAP experiments with the appropriate bioinformatic subtraction of the unbound fraction. This procedure greatly improved the quality of the dopaminergic neuron-specific (mDA) expression data and removed the majority of non-mDA genes likely coming from the surrounding glia cells. The described RNA expression approaches, that include single cell RNA sequencing to validate cell purity followed by bulk RNA sequencing, have been used to establish comprehensive mDA and serotonergic (SER) neuron-specific gene expression data. To address how these mDA and SER specific gene expression pattern are correlated with specific chromatin modifications (H3K4me3, H3K9me3, and H3K27me3) during developmental transitions and in the adult, the authors performed low-input chromatin immunoprecipitation followed by genome wide sequencing. In summary, the authors found a significant correlation between the presence of H3K27me3 and H3K27me3/H3K9me3 at genes that are repressed in adult mDA and SER neurons, and these suppressive chromatin modifications are inversely correlated with H3K4me3 and active gene expression. Next, to address if mDA genes associated with repressive chromatin modifications can be induced in response to cellular stresses, the authors perform mDA neuron-specific gene expression analysis after treatment with two different neurotoxins. The authors claim that H3K4me3+/H3K27me3+ bivalent loci in mDA are particularly “primed” to respond to cellular stressors such as 6-OHDA or methamphetamine (Figure 6). Using the mDA neuron specific TRAP approach, the authors identify 87 genes that are induced in mDA after exposure to the neurotoxic 6-OHDA injection. To identify truly bivalent K4me3/K27me3 genes in mDA -in contrast to genes that are differentially expressed within the mDA neuron population- the authors first excluded all genes that showed significant RNA expression in mDAs. Using this approach, the authors identified ~800 truly bivalent genes -among the original 2520 genes associated with both H3K4me3 and H3K27me3- that are transcriptionally silent in mDA neurons (Suppl. Table 5). Notably, this number of bivalent genes is similar to previously published data in adult neurons in the striatum. The conclusion that neurodegeneration-inducing cellular stress, such as 6-OHDA, leads to preferential induction of bivalent genes, however, is not supported by the data shown in Fig 6d. While the authors show a significant enrichment for mDA genes associated with H3K4me3/H3K27me3 among the 87 genes upregulated by 6-OHDA treatment, only 3/22 genes (Hrk, CD14, Mafb) (Suppl. Table 6) are truly bivalent genes based on the authors classification (Suppl. Table 5). The remaining genes are transcriptionally active and likely reflect differential gene expression and associated chromatin modification within the mDA cell population. Without these data supporting the physiological importance of “primed” bivalent genes in response to stress, the manuscript, while technically elegant and greatly improved, remains descriptive.

We appreciate the concern of the reviewer and have toned down the conclusions regarding de-repression of genes with bivalent promoters. It should be noted that the focus of our manuscript is not the possible de-repression of genes with bivalent promoters.

- *In the abstract on line 38 we have changed “de-repression” to “increased expression”*

- *The Subheading for the paragraph (line 407) describing the 6-OHDA and methamphetamine experiments has been changed from “De-repression of H3K27me3 marked genes during stress” to “Activation of H3K27me3 marked genes during stress”.*
- *In the discussion We have removed the following paragraph: “It has previously been shown that H3K27me3 marked genes can be de-repressed as a result of abnormal dopamine signaling in striatal MSNs. Our data show that in mDA-neurons, there was enrichment for H3K4me3/H3K27me3 bivalently marked genes that were activated by severe stress, including late-stage neurodegeneration and after systemic injection of methamphetamine (Figure 6). Two recent studies have suggested that the function of PRC2 in adult cells is to suppress transcription of bivalent genes while repression of the majority of H3K27me3-marked genes does not depend on PRC2 activity.”.*
- *On line 528 we have changed “de-repressed” to “were activated”.*
- *On line 531 we have changed “de-repressed” to “were activated”.*
- *At the end of the discussion we have removed the following paragraph: “Many K4/K27-marked genes up-regulated by 6-OHDA or methamphetamine were involved in normal processes, such as apoptosis and stress response. The induction of such genes cannot be considered aberrant under certain conditions. Nevertheless, they must be kept silenced in the absence of instructive signaling. One possibility is that the presence of H3K27me3 may serve as a threshold for gene induction for a subset of plasticity or apoptosis-related genes that can be overcome by transcription factor activation. Interestingly, in a recent study it was shown that upon loss of PRC2 activity in MSNs, bivalent genes were enriched among de-repressed whereas genes harboring H3K27me3 without H3K4me3 remained largely repressed. In addition, even though there was an induction of non-MSN transcriptional regulators, there was no pronounced transdifferentiation into other neuronal subtypes. This implies that in MSNs, alternative gene programs are repressed by additional mechanisms, and/or that the activation of non-MSN transcriptional regulators lacked the specificity to fully induce acquisition of an alternative cellular identity. It remains to be investigated if loss of PRC2-activity would have the same consequences for gene expression and phenotypic stability in mDA or SER-neurons.”*
- *Instead we have added: “Although a significant subset of promoter regions belonging to the K4/K27 state in NPCs, mDA and SER likely correspond to true promoter bivalency, future studies including conditional ablation of PRC2-components and sequential ChIP experiments, may clarify the extent and contribution of truly bivalent promoters to stress response and maintenance of cell type specific gene expression. Furthermore, such an effort would reveal if PRC2-associated gene silencing is indeed required for maintenance of cellular identity in these cell types.”*
- *In the title of figure legend for Figure 6 we have changed “De-repression” to activation.*

Response to reviewer #3: The manuscript by Sodersten, et al. is both important and well-done. They have addressed my concerns and I agree with their assertion that some other reviewer comments are beyond the scope of this study. One comment that I suggest for final revisions is that the authors recognize that while cell state in neurons appear permanent, cell type changes have been reported. Thus, I suggest a qualifier “ . . . largely retain their identity” be used in text. I recommend publication of the manuscript at this point.

- In the abstract, we have added “largely” as requested by the reviewer (line 29).